# Stealthy Yet Effective: Distribution-Preserving Backdoor Attacks on Graph Classification

Xiaobao Wang[1,2]   Ruoxiao Sun[1]   Yujun Zhang[1]   Bingdao Feng[1]
Dongxiao He[1]   Luzhi Wang[3]   Di Jin[1,*]

[1]College of Intelligence and Computing, Tianjin University, Tianjin, China
[2]Guangdong Laboratory of Artificial Intelligence and Digital Economy (SZ), Shenzhen, China
[3]College of Artificial Intelligence, Dalian Maritime University, Dalian, China
{wangxiaobao, rxiao_sun, zhangyujun}@tju.edu.cn
{fengbingdao, hedongxiao, jindi}@tju.edu.cn, wangluzhi0@gmail.com

## Abstract

Graph Neural Networks (GNNs) have demonstrated strong performance across tasks such as node classification, link prediction, and graph classification, but remain vulnerable to backdoor attacks that implant imperceptible triggers during training to control predictions. While node-level attacks exploit local message passing, graph-level attacks face the harder challenge of manipulating global representations while maintaining stealth. We identify two main sources of anomaly in existing graph classification backdoor methods: structural deviation from rare subgraph triggers and semantic deviation caused by label flipping, both of which make poisoned graphs easily detectable by anomaly detection models. To address this, we propose DPSBA, a clean-label backdoor framework that learns in-distribution triggers via adversarial training guided by anomaly-aware discriminators. DPSBA effectively suppresses both structural and semantic anomalies, achieving high attack success while significantly improving stealth. Extensive experiments on real-world datasets validate that DPSBA achieves a superior balance between effectiveness and detectability compared to state-of-the-art baselines. The code is available at https://github.com/TheCoderOfs/DPSBA.

## 1 Introduction

Graph Neural Networks (GNNs) are foundational models for learning on graph-structured data, achieving strong performance in tasks like node classification, link prediction, and graph classification [1]. Through message passing, GNNs capture structural and feature dependencies, enabling applications in social networks [2], recommender systems [3], and molecular analysis [4]. As GNNs are increasingly adopted in real-world systems, their security has drawn growing attention. Among various threats, backdoor attacks, where imperceptible triggers are embedded [5] in training data to manipulate predictions, pose a serious risk due to their stealth [6–8]. While extensively studied in vision and NLP, backdoor vulnerabilities in graph learning remain under exploration.

Most existing graph backdoor studies focus on node classification, where local triggers are injected to misclassify specific nodes [9, 10]. These attacks leverage GNN locality to propagate triggers through neighborhoods while remaining hard to detect. Link prediction attacks similarly operate in a local context, targeting the presence or absence of edges between node pairs [11]. In contrast, graph classification poses a fundamentally different and more complex challenge. Here, the attacker must influence the global semantics of an entire graph rather than a single node or edge. This requires

---

*Corresponding author

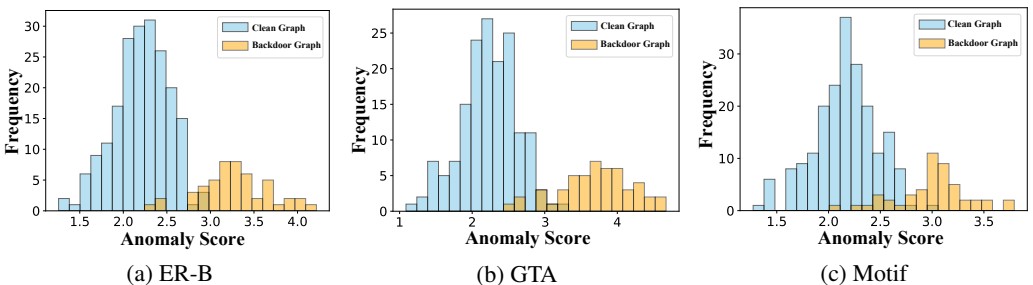

| (a) ER-B | (b) GTA | (c) Motif |

Figure 1: **Anomaly score distributions of clean vs. backdoor graphs on the AIDS dataset**. Anomaly scores are computed via SIGNET [12] for backdoor samples generated by ER-B [13], GTA [14], and Motif [15]. In all cases, backdoor graphs (yellow) shift significantly rightward compared to clean graphs (blue), revealing strong distributional deviation. This underscores the detectability of prior triggers and the need for distribution-preserving attacks.

triggers to manipulate the full-graph embedding, often through larger or structurally rare subgraphs. Recent backdoor attacks on graph classification attempt to address global manipulation challenges by injecting structurally distinct subgraphs, such as rare motifs [15]. However, these approaches often introduce obvious out-of-distribution (OOD) artifacts, which significantly compromise stealth and limit their practicality in real-world settings. As shown in Figure 1, there exists a clear distribution gap between clean and backdoor graphs, making current methods highly detectable by outlier detection models. Although Zhang et al. [16] similarly observe OOD issues in node classification and propose a distribution-preserving strategy, their approach relies on local perturbations that affect only a single node's embedding. In contrast, graph classification requires modifying the global message-passing process to influence the entire graph representation, making distribution preservation significantly more challenging (see Appendix C.1 for theoretical justification).

This global nature further amplifies the risk of both structural and semantic deviations when inserting backdoor triggers, as detailed in Appendix A: 1) **Structural Deviation**: Triggered by the injection of rare or unnatural subgraphs (e.g., low-frequency motifs) that diverge from the structural distribution of clean graphs. These triggers often create shortcut correlations with target labels and are easily identified by outlier detectors, as shown in Table A5. Rare motifs as triggers (Motif) achieve high attack success rates (ASR) but are highly detectable. In contrast, more frequent motifs (Motif-S) reduce detectability but also weaken attack effectiveness; 2) **Semantic Deviation**: Caused by label flipping, this introduces a discrepancy between a graph's assigned class and its inherent structure. As demonstrated in Table A6, adopting clean-label settings reduces the overall anomaly degree (AUC) across all methods but results in a notable ASR drop, revealing a fundamental trade-off between attack stealth and success. These findings highlight a critical limitation of existing methods: their inability to balance effectiveness and stealth. This raises a key challenge: *Can we design a graph-level backdoor attack that preserves the distributional properties of clean samples, avoids label manipulation, and remains both effective and stealthy?*

To tackle these challenges, we propose DPSBA (Distribution-Preserving Stealthy Backdoor Attack), a clean-label backdoor framework for graph classification. Unlike prior methods that use rare or manually designed subgraph triggers, often causing high anomaly scores, DPSBA adaptively learns in-distribution triggers from target-class data. To address both structural and semantic deviations, it introduces a joint training strategy combining: (1) a clean-label attack mechanism that preserves semantic consistency by avoiding label flipping, and (2) distribution-aware discriminators that penalize structural and feature-level anomalies. Through adversarial training with a surrogate classifier and anomaly detectors, DPSBA enables effective and stealthy trigger generation. Experiments on real-world datasets show that DPSBA substantially lowers anomaly scores while maintaining high attack success rates, achieving a strong trade-off between stealth and efficacy.

## 2 Preliminaries

In this section, we first give the definition of the graph classification task and then introduce our stealthy backdoor attack objective.

## 2.1 Graph Classification Task

Given a dataset consisting of graph-structured instances, the objective of graph classification is to assign a class label to each graph. Formally, a graph classification dataset is defined as: $\mathcal{C} = \{(G_1, y_1), (G_2, y_2), ..., (G_n, y_n)\}$, where each graph $G_i = (V_i, E_i)$ is composed of a set of nodes $V_i$ and edges $E_i$, and $y_i \in \mathcal{Y}$ is the corresponding ground-truth class label. Let $n$ denote the number of graph samples in the dataset. The goal is to learn a graph-level classifier: $f : \mathcal{G} \rightarrow \mathcal{Y}$, which maps each input graph $G \in \mathcal{G}$ to one of the class labels in $\mathcal{Y} = \{y_1, y_2, ..., y_n\}$. The classifier $f$ is typically trained via gradient-based optimization using a supervised loss function $\mathcal{L}_{train}$ (e.g., cross-entropy) over the training set $\mathcal{C}_{\text{train}}$.

## 2.2 Stealthy Backdoor Attack on Graph Classification Task

Backdoor attacks inject a hidden and malicious trigger mechanism (i.e., "backdoor") into the target model in advance. When the trigger appears, the backdoor is activated and misleads the model to get the desired output.

**Attacker's Goal.** The objective of a backdoor attacker in graph classification is threefold: (1) The model trained with injected backdoors should predict any graph containing the trigger as the attacker-specified target label $y_t$; (2) The model's performance on clean graphs should remain unaffected; (3) The backdoor graphs should not exhibit significant distributional deviations from clean graphs, ensuring high stealthiness. Formally, the attacker's goal can be formulated as:

$$f_{\theta_t}(m(G; g_t)) = y_t, \quad f_{\theta_t}(G) = f_\theta(G), \quad \text{diff}(f_o(m(G; g_t)), f_o(G)) \leq \epsilon, \tag{1}$$

where $G$ denotes a clean graph, and $y_t$ is the target label specified by the attacker. The function $m(G; g_t)$ denotes the trigger injection process, producing a backdoor graph $G_{g_t}$ by embedding trigger subgraph $g_t$ into $G$. $f_\theta$ and $f_{\theta_t}$ represent the clean and poisoned graph classification models, respectively. $f_o$ is a detection model trained on clean graphs, and $\text{diff}(\cdot, \cdot)$ measures the anomaly difference (e.g., anomaly score) between a clean graph and its backdoor counterpart. The scalar $\epsilon$ defines an upper bound on acceptable deviation for stealthiness, and can be tuned per dataset.

**Attacker's Knowledge and Capability.** We consider a restricted and realistic threat model in which the attacker has limited access and control: (1) **Black-box knowledge**: The attacker has no knowledge of the target model architecture, training hyperparameters, or optimization pipeline. This accommodates the possibility that different graph classification models may be deployed. (2) **Limited poisoning capability**: The attacker can only poison a small fraction of the training dataset and is not allowed to modify any labels. This reflects a more challenging clean-label setting and simulates real-world scenarios where data integrity is partially enforced.

## 3 Methodology

Grounded in the attacker's goal, DPSBA adaptively generates subgraph triggers whose topology and features mimic in-distribution patterns of clean graphs, reducing detectability while ensuring attack success. As shown in Figure 2, DPSBA includes two stages: poisoned sample construction and trigger optimization. In the first stage, hard samples from the target class are selected and injected with triggers at strategically chosen locations. The trigger's structure and features are generated by a learnable topology-feature generator. In the second stage, DPSBA jointly optimizes attack success via a surrogate model and minimizes detectability via anomaly-aware discriminators through staged adversarial training.

### 3.1 Poisoned Sample Construction

To support clean-label stealthy backdoor injection, DPSBA constructs poisoned samples from the target class itself, avoiding explicit label flipping. This process comprises three stages: (i) selecting informative samples for poisoning, (ii) identifying optimal injection locations within each graph, and (iii) initializing and injecting the trigger subgraph. Together, these steps ensure that the backdoor signal is both semantically consistent and statistically in-distribution.

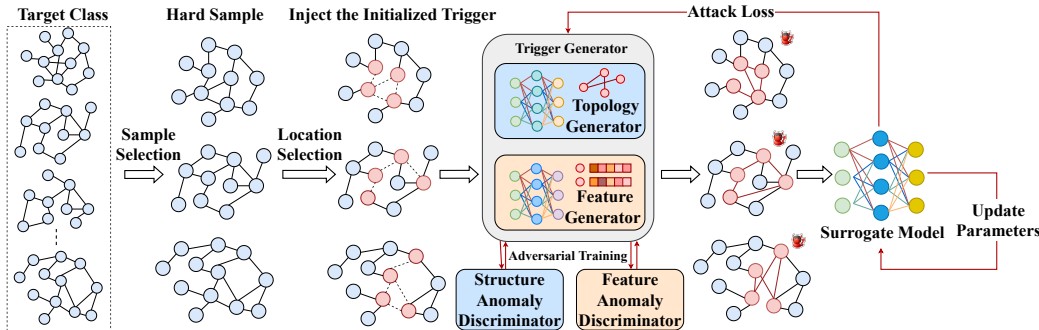

Figure 2: The overall structure of DPSBA

### 3.1.1 Hard Sample Selection

In the clean-label setting, directly selecting target-class samples avoids semantic inconsistency but may reduce the effectiveness of the attack. To counteract this, DPSBA strategically mines *hard samples*, i.e., samples from the target class that the model finds uncertain. This design is inspired by Gao et al. [17], who show that hard samples can amplify the effectiveness of backdoor attacks by being more susceptible to manipulation. In particular, low-confidence samples, those near the model's decision boundary, require smaller perturbations to shift predictions, and are thus better candidates for subtle backdoor injection. We use a clean surrogate model $f_\theta$ to measure the confidence score of a graph $G$ with respect to the target label $y_t$ as:

$$\mathrm{cfd}(G) = \mathrm{softmax}(f_\theta(G))_{y_t} = \frac{e^{f_\theta(G)_{y_t}}}{\sum_{j=1}^{K} e^{f_\theta(G)_j}}, \tag{2}$$

where $f_\theta(G) \in \mathbb{R}^K$ denotes the logit output of the surrogate model for graph $G$, $K$ is the total number of classes, and $y_t$ is the target class index. We select the bottom $p\%$ of target-class graphs with the lowest $\mathrm{cfd}(G)$ scores as poisoned samples. These hard samples are passed to the subsequent trigger location and injection stages, where subtle but effective perturbations are applied. Since they are more vulnerable to decision changes, they are more likely to be misclassified into the target class once the trigger is activated, while still maintaining clean-label consistency and high stealth.

### 3.1.2 Trigger Location Selection

After selecting the hard target-class samples for poisoning, DPSBA determines where in each graph to inject the trigger. Existing studies adopt various strategies to identify influential nodes, such as using centrality metrics [15] or extracting interpretable subgraphs [18, 19]. While interpretation-based methods offer precision, they often incur prohibitive computational costs. To strike a balance between scalability and informativeness, DPSBA adopts a two-stage selection process. First, nodes with high degree centrality are pre-selected as candidates. Specifically, for a graph $G$ with $N$ nodes, we compute the normalized degree centrality of each node $v$ as $\deg(v)/(N-1)$ and retain the top-$k$ nodes (with $k$ typically set to $2M$, where $M$ is the trigger size). Next, among these candidates, we identify the $M$ most influential nodes using a surrogate-based ablation approach. For each node $v$ in the candidate set $\mathcal{N}_{can}$, we evaluate its contribution to model prediction by measuring the change in output after removing it:

$$S(v) = |f_\theta(G + \Delta_v) - f_\theta(G)|, \tag{3}$$

where $\Delta_v$ denotes the removal of node $v$ from $G$. The top-$M$ nodes with the highest $S(v)$ values are selected as trigger attachment nodes. These nodes are then replaced, along with their edges, by an initialized subgraph trigger $g_t$ to prepare for the subsequent generation phase.

### 3.1.3 Trigger Generation and Injection

To enable the insertion of stealthy and effective triggers, generative approaches offer superior flexibility and control over random, interpretability-based, or search-based strategies. Inspired by GTA [14], DPSBA introduces two lightweight yet expressive modules, i.e., a **topology generator** and a **feature generator**, that cooperatively define the structure and node attributes of the injected subgraph. These generators are trained in a separate phase to adaptively fit the data distribution.

**Topology Generator.** The topology generator is implemented as a multilayer perceptron (MLP) that transforms the adjacency matrix $\mathbf{H}$ of the target injection region into a learnable soft structure: $\mathbf{H}' = \sigma(W_1\mathbf{H} + b_1)$, where $W_1$ and $b_1$ are learnable parameters (for the convenience of subsequent discussion, they will be simply referred to as $\omega_t$) and $\sigma(\cdot)$ is a non-linear activation function. For undirected graphs, the adjacency matrix is symmetrized as $\mathbf{A} = \frac{1}{2}\left(\mathbf{H}' + \mathbf{H}'^\top\right)$. To accommodate the discrete nature of graph topology, we adopt a binarization mechanism inspired by binarized neural networks [20], where the final adjacency matrix is given by: $\mathbf{A}_{binary} = \mathbb{I}(\mathbf{A} > 0.5)$. Binarization is applied only during forward propagation to preserve gradient flow during backpropagation. The resulting binary matrix defines the trigger's structural connectivity.

**Feature Generator.** To ensure the feature space consistency between clean and poisoned graphs, the feature generator produces trigger node features based on the original features at the injection site. It employs an MLP to generate: $\mathbf{X}' = \sigma(W_2\mathbf{X} + b_2)$, where $\mathbf{X}$ denotes the original features of the injected nodes, and $W_2$, $b_2$ are learnable parameters, which are simply referred to as $\omega_f$ for the convenience of subsequent discussion. This transformation ensures that generated features remain aligned with the data distribution, thereby reducing feature-level anomaly. Besides, the feature generator can also avoid distribution shift when the target graphs have high attribute variance. The feature generator does not rely on distribution priors of the entire dataset. Instead, it takes as input the local structural and attribute context around the trigger injection site and generates features that blend smoothly with the surrounding neighborhood. This allows the generator to adapt to the specific variance of each target graph on a per-instance basis, without requiring any handcrafted normalization or explicit regularization. As a result, the generated features remain close to the local data manifold, reducing the likelihood of creating detectable anomalies, even in the presence of high attribute variance across graphs.

**Trigger Injection.** Given a selected clean graph $G$ and the generated trigger subgraph $g_t = (\mathbf{A}_{binary}, \mathbf{X}')$, DPSBA injects the trigger at the designated node locations to produce a poisoned sample: $G_{g_t} = m(G; g_t)$, where $m(\cdot)$ denotes the injection operation. The resulting $G_{g_t}$ is added to the training set along with clean graphs. After the backdoored model is trained, the attacker can craft adaptive triggers during inference and inject them into arbitrary test graphs to activate the backdoor and induce targeted misclassification.

### 3.2 Trigger Optimization

Once the trigger subgraph is initialized, DPSBA enters the optimization phase to jointly refine the topology and feature generators for both effectiveness and stealth. It minimizes a hybrid objective comprising: (1) an attack loss that enforces confident misclassification into the target class, and (2) an adversarial anomaly loss that suppresses structural and feature-level deviations detectable by outlier models. This is achieved via a two-stage adversarial training strategy with dedicated discriminators and a dynamically updated surrogate model.

**Attack Effectiveness.** The core goal of a backdoor attack is to ensure that the trigger-embedded graph $G_{g_t}$ is classified into the target class $y_t$. This is achieved by minimizing the attack loss with respect to a surrogate model $f_{\theta^*}$:

$$\mathcal{L}_{atk} = -\log f_{\theta^*}(G_{g_t})_{y_t}, \tag{4}$$

where $f_{\theta^*}(\cdot)_{y_t}$ denotes the logit output of the surrogate model for class $y_t$.

**Stealthiness via Adversarial Anomaly Minimization.** To ensure the trigger remains statistically inconspicuous, DPSBA introduces adversarial discriminators trained to distinguish clean and backdoor samples from structure and feature perspectives. Specifically, the topology discriminator $D_{\theta_t}$ (a GCN) detects structural anomalies, while the feature discriminator $D_{\theta_f}$ (an MLP) detects feature distribution shifts. The generators aim to fool these discriminators via the following minimax objectives:

$$\min_{\omega_t}\max_{\theta_t}\mathcal{L}_d^{(t)} = \sum_{G\sim\mathcal{G}_c}\log D_{\theta_t}(G) + \sum_{G\sim\mathcal{G}_b}\log(1 - D_{\theta_t}(G_{g_t}(\omega_t))), \tag{5}$$

$$\min_{\omega_f}\max_{\theta_f}\mathcal{L}_d^{(f)} = \sum_{G\sim\mathcal{G}_c}\log D_{\theta_f}(G) + \sum_{G\sim\mathcal{G}_b}\log(1 - D_{\theta_f}(G_{g_t}(\omega_f))), \tag{6}$$

where $\mathcal{G}_c$ and $\mathcal{G}_b$ are the clean and poisoned graph sets, and $\omega_t$, $\omega_f$ are the generator parameters.

**Joint Training Objectives.** DPSBA jointly optimizes the generators using a weighted sum of the attack loss and the corresponding adversarial loss. For the topology generator:

$$\min_{\omega_t} \sum_{G \in \mathcal{G}_b} \mathcal{L}_{atk}(G_{g_t}(\omega_t)) + \alpha \mathcal{L}_d^{(t)}(D_{\theta_t}(G_{g_t}(\omega_t))), \quad \text{s.t. } \theta^* = \arg\min_{\theta} \mathcal{L}_{train}(f_\theta(C)), \quad (7)$$

and for the feature generator:

$$\min_{\omega_f} \sum_{G \in \mathcal{G}_b} \mathcal{L}_{atk}(G_{g_t}(\omega_f)) + \beta \mathcal{L}_d^{(f)}(D_{\theta_f}(G_{g_t}(\omega_f))), \quad \text{s.t. } \theta^* = \arg\min_{\theta} \mathcal{L}_{train}(f_\theta(C)), \quad (8)$$

where $\alpha$ and $\beta$ are hyperparameters balancing stealth and attack objectives, and $C$ is the clean training set used to update the surrogate model.

**Adversarial Training Strategy.** To realize both stealth and attack effectiveness, DPSBA adopts a staged adversarial training strategy to optimize the topology and feature generators in coordination with anomaly discriminators. The joint objective encourages trigger-embedded graphs to be confidently misclassified into the target class while remaining statistically similar to clean samples. To further justify this design, we derive in Appendix C.2 a formal lower bound that connects the total variation distance between clean and poisoned graph distributions with the optimal anomaly detection AUC, demonstrating that minimizing such divergence directly improves stealthiness.

The training process consists of two alternating phases: **1) Topology optimization phase:** The topology generator $\omega_t$ and discriminator $D_{\theta_t}$ are alternately updated. The generator learns to create subgraph structures that both maximize attack loss $\mathcal{L}_{atk}$ and minimize structural anomaly signals detectable by $D_{\theta_t}$. **2) Feature optimization phase:** Similarly, the feature generator $\omega_f$ and discriminator $D_{\theta_f}$ are jointly trained. The generator adapts node features to support effective attacks while suppressing attribute-level anomalies identified by $D_{\theta_f}$. To maintain alignment with the evolving trigger distribution, the surrogate model $f_\theta$ is fine-tuned after each phase, ensuring reliable attack gradient signals throughout training. This cooperative optimization framework allows DPSBA to dynamically balance the trade-off between attack success and stealth, as further summarized in Appendix D. Time complexity analysis can be found in Appendix F.

# 4 Experiment

We first introduce the experimental setup, including datasets, baselines, and implementation details. Then, we evaluate DPSBA's performance on attack effectiveness and stealth across multiple settings, followed by ablation analysis and anomaly stealth evaluation. Additional hyperparameter studies are included in Appendix E.4 E.5.

## 4.1 Experimental Setup

### 4.1.1 Datasets and Evaluation Metrics

We evaluate DPSBA on four real-world graph classification datasets from the TUDataset benchmark [21]: **PROTEINS_full** [22] ( protein graphs for function prediction), **AIDS** [23] (molecular graphs related to AIDS research), **FRANKENSTEIN** [24] (a compound property dataset combining BURS and MNIST features), and **ENZYMES** [25, 26] (a 6-class biomolecular classification task). For each dataset, we designate the minority class as the attack target to simulate a realistic low-frequency scenario. The statistics for the datasets are summarized in Table E7. We adopt three metrics: **ASR** (Attack Success Rate) for attack effectiveness, **CAD** (Clean Accuracy Drop) to assess model utility degradation, and **AUC** (Area Under the ROC Curve) to quantify anomaly detectability via outlier models. Detailed metric definitions are deferred to Appendix E.2.

### 4.1.2 Baselines

We compare DPSBA with five representative graph-level backdoor attack methods. **ER-B** [13] generates universal triggers using the Erdős–Rényi random graph model. **LIA** [18] modifies the connections of low-importance nodes, as identified by GNN explanation techniques, with fixed trigger structures. **GTA** [14] employs a bi-level optimization strategy to train a subgraph generator that adapts

triggers per graph. **Motif** [15] selects low-frequency motifs from the dataset as effective but easily detectable triggers. **Motif-S** is a stealthier variant using the high-frequency M41 motif, reducing anomaly scores while slightly sacrificing attack strength. Following GTA [14], we split each dataset into 50% training and 50% test sets, with 5% of the training data poisoned. To ensure fair comparison, all methods adopt a fixed trigger size of 4. Both the topology and feature generators are trained for 20 epochs per stage over 3 iterations with a learning rate of 0.001, using early stopping [27]. For baselines, we use the best hyperparameters reported in their original papers.

### 4.1.3 Graph Classification Models and Anomaly Detection Algorithms

Following prior work on graph backdoor attacks [15], we evaluate DPSBA on three widely used graph classifiers: GCN [28], GIN [29], and SAGPool [30]. These models are used both as attack targets and surrogates to test generalizability across architectures. Clean accuracy results are reported in Table E8. For anomaly detection, we adopt SIGNET [12], identified as the most effective method in a recent benchmark study [31], to assess the stealthiness of injected backdoor samples.

## 4.2 Experimental Results

Table 1: Comparison results between DPSBA and each baseline model

| Datasets | Surrogate Model | Metrics | ER-B | LIA | GTA | Motif | Motif-S | Ours |
|---|---|---|---|---|---|---|---|---|
| PROTEINS_full | GCN | ASR (%) | 51.53 | 68.35 | 73.16 | 70.91 | 48.56 | **73.93** |
| | | CAD (%) | 4.73 | 4.70 | 5.14 | 5.92 | 4.66 | **4.62** |
| | | AUC (%) | 70.04 | 71.01 | 78.20 | 79.16 | 64.72 | **60.11** |
| | GIN | ASR (%) | 62.53 | 58.77 | 80.96 | 79.08 | 63.01 | **87.91** |
| | | CAD (%) | 4.88 | 4.36 | 4.57 | 4.97 | **4.33** | 4.92 |
| | | AUC (%) | 79.65 | 71.74 | 79.96 | 80.06 | 70.49 | **62.95** |
| | SAGPool | ASR (%) | 65.38 | 64.81 | 94.04 | 71.35 | 57.09 | **94.15** |
| | | CAD (%) | 4.26 | 5.02 | 3.65 | 3.36 | 3.94 | **3.29** |
| | | AUC (%) | 71.34 | 76.89 | 78.57 | 82.75 | 81.81 | **69.20** |
| AIDS | GCN | ASR (%) | 85.38 | 85.49 | 93.21 | 92.69 | 56.08 | **94.76** |
| | | CAD (%) | 4.53 | 3.80 | 5.14 | 4.12 | 4.03 | **2.38** |
| | | AUC (%) | 98.08 | 97.22 | 99.34 | 99.71 | 89.43 | **72.65** |
| | GIN | ASR (%) | 93.99 | 95.56 | 97.52 | **97.75** | 56.8 | 95.87 |
| | | CAD (%) | 2.69 | 2.03 | 2.65 | 2.28 | 2.51 | **1.94** |
| | | AUC (%) | 99.98 | 99.20 | 99.34 | 99.71 | 94.29 | **73.66** |
| | SAGPool | ASR (%) | 59.26 | 62.66 | 86.99 | 87.65 | 62.89 | **98.90** |
| | | CAD (%) | 1.65 | 1.79 | 3.77 | 2.64 | 2.44 | **-0.40** |
| | | AUC (%) | 95.79 | 94.56 | 99.67 | 99.02 | 93.43 | **77.23** |
| FRANKEN-STEIN | GCN | ASR (%) | 63.60 | 61.04 | **99.35** | 80.57 | 59.24 | 98.37 |
| | | CAD (%) | 1.71 | 1.56 | 2.74 | 1.15 | 3.96 | **1.01** |
| | | AUC (%) | 80.41 | 75.66 | 100.00 | 89.64 | 69.23 | **68.96** |
| | GIN | ASR (%) | 92.06 | 82.63 | 98.65 | 92.87 | 58.68 | **99.84** |
| | | CAD (%) | 3.60 | 2.35 | 1.95 | 2.44 | 1.75 | **1.83** |
| | | AUC (%) | 85.73 | 76.15 | 91.06 | 87.54 | **65.77** | 73.46 |
| | SAGPool | ASR (%) | 68.15 | 90.18 | 95.23 | 84.56 | 52.29 | **99.99** |
| | | CAD (%) | 4.78 | 4.66 | 4.64 | 4.61 | 6.86 | **4.57** |
| | | AUC (%) | 64.89 | 77.50 | 80.46 | 87.29 | 60.98 | **60.12** |
| ENZYMES | GCN | ASR (%) | 26.09 | 30.43 | 95.33 | 21.74 | 15.21 | **96.67** |
| | | CAD (%) | 4.17 | 4.99 | 3.00 | 4.99 | **-1.67** | -0.67 |
| | | AUC (%) | 68.32 | 66.15 | 71.20 | 71.35 | 66.22 | **66.11** |
| | GIN | ASR (%) | 37.83 | 27.02 | 96.00 | 16.21 | 12.16 | **99.33** |
| | | CAD (%) | 9.17 | 10.00 | 2.67 | 8.33 | 4.17 | **-0.33** |
| | | AUC (%) | 71.40 | 62.01 | 76.42 | 68.18 | 65.78 | **41.20** |
| | SAGPool | ASR (%) | 29.54 | 38.63 | 100.00 | 15.91 | 11.37 | **100.00** |
| | | CAD (%) | 4.33 | 6.67 | 5.00 | 10.83 | **3.33** | 4.00 |
| | | AUC (%) | 57.73 | 63.98 | 70.37 | 75.47 | 69.48 | **49.91** |

Table 2: Results of the transferability evaluation(%)

| Surrogate model | Actual model | PROTEINS_full | | AIDS | | FRANKENSTEIN | |
|---|---|---|---|---|---|---|---|
| | | ASR | CAD | ASR | CAD | ASR | CAD |
| GCN | GIN | 81.32 | 4.79 | 99.44 | 1.01 | 98.37 | 0.03 |
| | SAGPool | 98.90 | 0.08 | 96.14 | 2.48 | 94.96 | -0.10 |

### 4.2.1 Effectiveness and Stealthiness

We evaluate DPSBA on four datasets against six baselines in terms of attack effectiveness (ASR), model performance drop (CAD), and anomaly detectability (AUC), as shown in Table 1. DPSBA consistently achieves optimal or near-optimal ASR across datasets and models, verifying its strong attack capability. This is credited to its adaptive trigger generation guided by confident hard samples and structurally important nodes. In terms of CAD, DPSBA maintains minimal accuracy degradation (<5%), indicating high stealth at the model level. For distribution-level stealth, DPSBA significantly reduces AUC, i.e., maintaining values around 70%, demonstrating strong resistance to detection by statistical outlier models. We further observe that ASR and AUC fluctuate significantly across datasets due to inherent differences in graph structure and anomaly sensitivity. For datasets particularly sensitive to attribute or structural perturbations (e.g., FRANKENSTEIN), even minor modifications can induce high ASR yet large anomaly scores. In such cases, we recommend an early stopping strategy, namely terminating generator updates once ASR saturates or marginal gains diminish, to enhance stealth. For instance, on FRANKENSTEIN, where attribute anomalies are highly detectable (GTA achieves >95% ASR but with high AUC), DPSBA disables the feature generator early, achieving near-perfect ASR solely via structural triggers. Across all benchmarks, DPSBA achieves the high ASR of Motif while matching the stealth of Motif-S, validating its ability to balance effectiveness and detectability. For multi-class graph classification tasks, DPSBA consistently outperforms all baselines in terms of both attack success rate (ASR) and stealth (CAD, AUC) across three surrogate models. Notably, most baselines (except GTA) fail on this dataset due to high attribute and strucutre variability, underscoring the advantage of our distribution-preserving and adaptive design.

### 4.2.2 Transferability

To evaluate the transferability of DPSBA, we use GCN as the surrogate model to generate triggers and test them on other classifiers. Table 2 reports ASR and CAD across datasets using GIN and SAGPool as target models. Since AUC is model-independent, it is omitted. Results show that DPSBA maintains high ASR and low CAD across architectures, demonstrating strong transferability even when using a simple surrogate model. Meanwhile, we observe a counterintuitive phenomenon. A comparison of the experimental results on the AIDS and PROTEINS_full datasets in Table 1 and Table 2 reveals that the ASR is higher when the surrogate model differs from the actual model than when they are identical. We provide a detailed discussion of this matter in Appendix E.6.

### 4.2.3 Anomaly Stealth Analysis

To show the anomaly stealthiness of DPSBA, we first visualize the anomaly score distributions of clean and backdoor graphs on all datasets in Figure 3. The distributions show strong overlap, indicating that

Table 3: Performance (AUC%) of different anomaly detection models

| Models | PROTEINS_full | AIDS | FRANKENSTEIN |
|---|---|---|---|
| OCGIN | 55.19 | 86.52 | 72.12 |
| GLocalKD | 50.24 | 31.46 | 46.57 |
| SIGNET | 60.11 | 72.65 | 68.96 |

DPSBA-generated triggers exhibit minimal anomaly and are hard to distinguish from clean samples. This is largely attributed to the use of clean-label constraints and adaptive optimization. To further assess robustness across detectors, we evaluate DPSBA under multiple anomaly detection algorithms, including OCGIN [32] and GLocalKD [33], with results summarized in Table 3. DPSBA consistently achieves low AUCs and, notably, GLocalKD fails to detect most triggers, confirming the strong stealth of DPSBA across detection methods. Beyond anomaly detection, we examine randomized subsampling [13, 14], a standard defense strategy that randomly removes subgraphs during training. As shown in Figure 4(a), DPSBA maintains high ASR despite this defense, thanks to its compact

trigger size and dual embedding in structure and features, making the trigger difficult to eliminate via sampling.

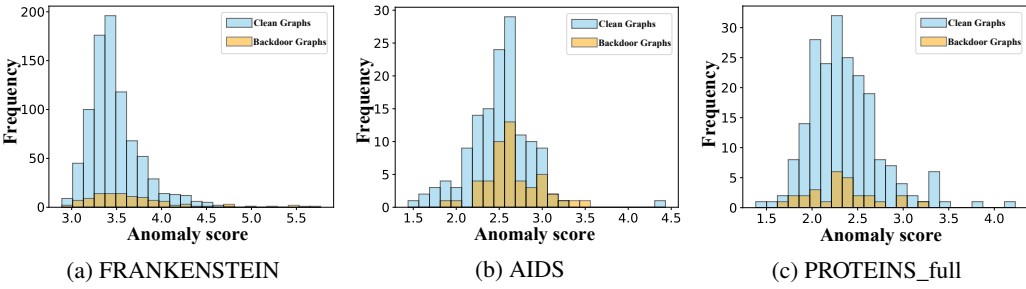

(a) FRANKENSTEIN  (b) AIDS  (c) PROTEINS_full

Figure 3: Anomaly distribution visualization

We also test anomaly-based training data filtering. Taking AIDS as an example, we use SIGNET to remove the top 5% anomalous samples before training. As Figure 4(b) shows, DPSBA remains robust, suffering minimal ASR drop even when most poisoned samples are detected. In contrast, previous methods show significant degradation. This highlights

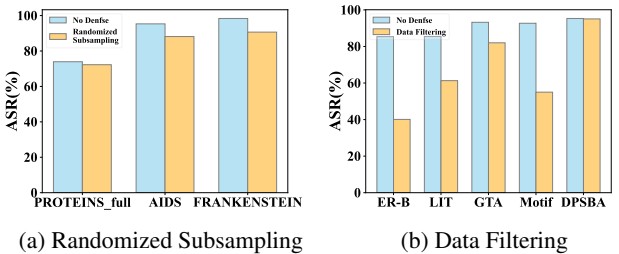

(a) Randomized Subsampling  (b) Data Filtering

Figure 4: Attack performance under defense

a key insight: existing detection strategies, while effective at identifying poisoned data, remain insufficient to fully prevent backdoor success due to the low sample requirement for trigger activation. More proactive and robust defenses are needed to counter stealthy attacks like DPSBA.

## 4.3 Ablation Study

To evaluate the role of each component in DPSBA, we conduct ablation experiments on PROTEINS_full and AIDS by removing one module at a time: DPSBA/S (w/o hard sample selection), DPSBA/N (w/o position selection), DPSBA/F (w/o feature generator), DPSBA/T (w/o topology generator), and DPSBA/OD (w/o adversarial

Table 4: Results of the ablation experiments (%)

| Model | PROTEINS_full | | | AIDS | | |
|---|---|---|---|---|---|---|
| | ASR | CAD | AUC | ASR | CAD | AUC |
| **DPSBA** | 73.93 | 4.62 | 60.11 | 94.76 | 2.38 | 72.65 |
| DPSBA/S | 70.98 | 3.57 | 60.24 | 91.32 | 2.09 | 72.60 |
| DPSBA/N | 70.74 | 4.53 | 58.97 | 93.67 | 2.31 | 71.26 |
| DPSBA/F | 71.80 | 4.96 | 59.01 | 85.67 | 2.40 | 67.26 |
| DPSBA/T | 69.08 | 3.71 | 54.73 | 93.66 | 2.91 | 71.41 |
| DPSBA/OD | 90.88 | 4.90 | 90.23 | 99.46 | 3.54 | 93.72 |

training), with results shown in Table 4. **Note:** The FRANKENSTEIN dataset is highly sensitive to attribute perturbations, where even small feature changes can lead to high ASR and pronounced anomaly. As discussed in Section 4.2.1, we apply early stopping to the feature generator on this dataset; thus, it is excluded from the ablation study.

Removing the sample or location selection module (S/N) slightly reduces ASR, confirming their effectiveness in identifying vulnerable graph regions. Excluding either generator (F/T) significantly degrades ASR but slightly improves AUC, suggesting that targeting both topology and features is essential for strong attacks, while single-aspect perturbation introduces fewer anomalies. The relative importance of these attributes varies across datasets, indicating that under strict stealth constraints, selectively optimizing one may be preferable. DPSBA/OD achieves the highest ASR but suffers the worst AUC, highlighting the importance of adversarial training for stealth. All other variants maintain CAD within 5%, showing minimal impact on model performance. Overall, the full DPSBA provides the best trade-off between effectiveness and stealth, with adversarial training enhancing detectability resistance and other modules improving attack strength.

## 4.4 Impact of the Loss Weights $\alpha$ and $\beta$

To control the anomaly level of injected triggers, DPSBA employs adversarial training with two loss weights: $\alpha$ for structural anomalies and $\beta$ for feature anomalies. This experiment investigates how varying these weights affects the trade-off between attack effectiveness and stealth. Specifically, we vary $\alpha$ and $\beta$ from 0.1 to 100 in the

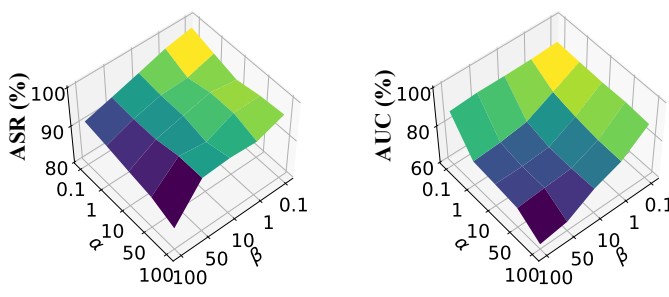

Figure 5: Parametric analysis of $\alpha$ and $\beta$

joint loss formulations (Equations 7 and 8) and observe the corresponding changes in attack success rate (ASR) and anomaly detectability (AUC). Results are shown in Figure 5. Overall, both ASR and AUC exhibit a downward trend as $\alpha$ and $\beta$ increase. Larger weights impose stronger anomaly constraints, limiting the model's aggressiveness and thereby enhancing stealth. However, the rate of decline differs between $\alpha$ and $\beta$, reflecting their distinct effects on structural and feature-based anomaly regularization, respectively. This suggests the two parameters can be tuned independently to suit dataset-specific characteristics. Importantly, when $\alpha$ and $\beta$ exceed a certain threshold, AUC stabilizes, indicating diminishing returns from further increasing anomaly penalties. This plateau signals that stealth constraints have been effectively enforced. At this point, DPSBA achieves a favorable balance between high ASR and low AUC. This analysis provides practical guidance for tuning $\alpha$ and $\beta$ based on application-specific requirements: small values favor stronger attacks, while larger values improve stealth, helping practitioners achieve optimal trade-offs on different datasets.

## 5 Conclusion

In this paper, we present **DPSBA**, a clean-label backdoor framework tailored for graph classification. Unlike prior work that relies on out-of-distribution triggers, semantic label flipping, or manually designed patterns, DPSBA learns in-distribution triggers directly from target-class graphs via a joint optimization process. The framework integrates a clean-label attack mechanism with anomaly-aware adversarial training, optimizing both attack effectiveness and stealth. Specifically, it employs structural and feature-level discriminators to minimize detectable deviations while preserving semantic consistency. Extensive experiments on diverse graph benchmarks and architectures demonstrate that DPSBA consistently achieves high attack success rates (ASR), low anomaly detection scores (AUC), and negligible clean accuracy drop (CAD). These results validate that DPSBA achieves a favorable balance between stealth and attack potency, outperforming existing baselines in both effectiveness and detectability. Our findings highlight the practicality of stealthy, distribution-preserving backdoor attacks in real-world graph learning scenarios.

**Limitations.** The clean-label constraint, while significantly enhancing stealth, may limit attack potency in scenarios with imbalanced class distributions or scarce target-class samples, where clean instances are insufficient to support effective trigger learning. Additionally, DPSBA operates under a partial training data access assumption, which restricts its applicability to fully black-box scenarios or test-time attacks where no data access is available. Future work may investigate lightweight or black-box-compatible variants of DPSBA, explore adaptation to broader graph settings such as heterophilic or dynamic graphs, and develop theoretically grounded defenses capable of detecting distribution-preserving triggers.

## 6 Acknowledgments

This work was supported by the National Natural Science Foundation of China (No. 62302333, No. 92370111, No. 62272340, No. 62422210, No. 62276187 and No. 62502065) , the Open Research Fund from Guangdong Laboratory of Artificial Intelligence and Digital Economy (SZ) (No. GML-KF-24-16), and Hebei Natural Science Foundation (No. F2024202047).

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

# A   Anomaly Issues in Graph Classification Attacks

To demonstrate that backdoor graphs in graph classification tasks tend to exhibit clear out-of-distribution (OOD) characteristics, we conduct anomaly detection experiments on the AIDS dataset [23]. Specifically, we use the SIGNET algorithm [12] to evaluate the statistical anomaly of poisoned samples generated by representative backdoor attack methods, including ER-B [13], LIA [18], GTA [14], and Motif [15], comparing them against clean training samples. As shown in Table A5, we adopt AUC as the evaluation metric for anomaly detection, which reflects how well a detector can distinguish clean graphs from anomalous ones. The results show that most existing methods result in extremely high AUC scores, often exceeding 90%, highlighting that backdoor samples are statistically distinguishable and exhibit significant anomalies. This stands in contrast to node-level backdoor attacks, where the GNN's local message passing can help diffuse trigger signals and reduce detectability.

Table A5: Performance of backdoor attack methods on the AIDS dataset

| Metrics | Backdoor Attack Method | | | | |
| --- | --- | --- | --- | --- | --- |
| | ER-B | LIA | GTA | Motif | Motif-S |
| ASR(%) | 85.38 | 85.49 | 93.21 | 92.69 | 56.08 |
| AUC(%) | 98.08 | 97.22 | 99.34 | 99.71 | 89.43 |
| CAD(%) | 4.53 | 3.80 | 5.14 | 4.12 | 4.03 |

**Structural Deviation.**   Backdoor models often exploit shortcut learning by associating rare subgraph triggers with the target class to enhance attack success. However, these rare triggers tend to deviate significantly from the natural graph distribution, making them easy to detect. Referring to the trigger selection strategy in Motif [15], we design a comparison between two variants: (1) Motif uses the least frequent motif as the trigger, while (2) Motif-S adopts the most common motif in the AIDS dataset. As shown in Table A5, Motif achieves higher ASR due to the strong signal of the rare trigger. However, this also results in extremely high AUC scores, indicating poor stealth. In contrast, Motif-S trades off some attack effectiveness for significantly improved stealth, as frequent motifs are naturally present in the dataset and thus harder for anomaly detectors to distinguish. This validates that trigger frequency directly influences the detectability of structural anomalies.

Table A6: Performance of clean label backdoor attack

| Metrics | Backdoor Attack Method | | | | |
| --- | --- | --- | --- | --- | --- |
| | ER-B | LIA | GTA | Motif | Motif-S |
| ASR(%) | 37.11 | 57.62 | 91.88 | 84.37 | 1.80 |
| AUC(%) | 61.03 | 59.42 | 98.05 | 97.74 | 71.88 |
| CAD(%) | 4.33 | 3.78 | 4.92 | 4.24 | 4.13 |

**Semantic Deviation.**   Label manipulation, commonly used in traditional backdoor attacks, introduces semantic inconsistency between the graph content and its assigned label, exacerbating anomaly. The success of such attacks relies on enforcing a co-occurrence between the trigger and target label, which is typically achieved by inserting the trigger into non-target-class samples and forcibly flipping their labels. This label tampering often results in category-level semantic conflict. To assess its effect, we further evaluate the same backdoor methods under a clean-label setting (i.e., without modifying labels). As shown in Table A6, while the AUC scores drop in all methods, indicating improved stealth, the ASR also drops significantly. This suggests that semantic consistency reduces the statistical footprint of the backdoor, but at the cost of attack success. In summary, label flipping enhances attack strength but induces additional semantic anomaly, making backdoor graphs more detectable.

These findings highlight two major anomaly sources in graph-level backdoor attacks: structural deviation from rare subgraphs and semantic inconsistency due to label tampering. This motivates the design of DPSBA, which aims to preserve distributional characteristics by avoiding both sources of anomaly.

# B  Related Work

## B.1  General Graph Backdoor Attacks

Backdoor attacks constitute a class of adversarial attacks in which an attacker implants a hidden behavior, known as a trigger mechanism, into the model during training. Once trained, the model behaves normally on clean inputs, maintaining high utility and stealth. However, when presented with an input containing a specific trigger pattern, the backdoor is activated and causes the model to misclassify the input into an attacker-specified target class. This property makes backdoor attacks particularly insidious in real-world deployments. In the graph domain, various works have explored backdoor threats in settings beyond standard graph classification. For node classification tasks, researchers have proposed attacks that inject malicious subgraphs or perturb node features to manipulate predictions. For example, Zhang et al.[34] first investigated the stealthiness dimension of backdoor attacks on node classification, while Wang et al.[35] introduced multi-target attacks to expand adversarial flexibility. Zhao et al.[36] proposed a spectral backdoor method by subtly altering the frequency domain of node features. For link prediction tasks, Zheng et al.[37] constructed triggers composed of fabricated nodes and target link pairs, whereas Dai et al.[11] showed that even single-node triggers can influence link predictions. Other studies have expanded backdoor threats into more specialized domains. For example, Zhang et al.[6] proposed the first attack on graph contrastive learning[38, 39], and Alrahis et al.[40] designed a hardware-level GNN backdoor mechanism. In the context of graph prompt learning, Lyu et al.[41] framed backdoor injection as a feature-collision optimization problem. Federated graph learning has also been explored: Xu et al. proposed two attack paradigms: distributed backdoor attack (DBA)[42] and Centralized Backdoor Attack (CBA)[43], targeting collaborative learning environments through data poisoning strategies. Although recent efforts have extended graph backdoor attacks to a wide range of scenarios, the graph classification task remains the primary focus due to its foundational role in bioinformatics, chemistry, and cybersecurity. Its holistic graph-level nature presents both unique challenges and opportunities for backdoor research.

## B.2  Backdoor Attacks against Graph Classification Tasks

Existing backdoor attack methods targeting graph classification can be broadly categorized into four groups based on their trigger generation strategies: (1) random pattern generation, (2) interpretability-based generation, (3) gradient-based optimization, and (4) distribution-aware or search-based methods. **Random Generation via Graph Distribution:** Zhang et al.[13] first introduced a backdoor attack against GNNs by randomly generating subgraphs using Erdős–Rényi (ER) models and inserting them as universal triggers into the training set. This demonstrated that fixed subgraph patterns can reliably activate backdoors in graph classification. Sheng et al.[44] further incorporated statistical characteristics of the dataset to create hybrid local-global trigger patterns, improving both the flexibility and efficacy of the attack. **Explainability-Based Trigger Design:** Xu et al.[18] leveraged GNNExplainer to assess the importance of the node and identify the optimal injection sites, the least important or the most influential nodes, for the trigger subgraphs. This method emphasized the influence of structural semantics on model decisions and improved the stealthiness of trigger placement. Building on this idea, Wang et al.[19] proposed inserting explainable subgraphs as triggers. Tong et al.[45] took this further by identifying and replacing predictive substructures in graphs to construct semantically-aligned triggers. **Gradient-Based Optimization Methods:** Xi et al.[14] proposed GTA, a general and dynamic backdoor attack framework that learns subgraph triggers via bilevel optimization. Unlike fixed-pattern methods, GTA can adapt trigger structures to specific graph instances, enhancing attack flexibility and stealth. Similarly, Yang et al.[46] developed TRAP, which perturbs graphs through gradient-driven strategies to generate transferable, pattern-free backdoor triggers. **Search-Based or Distribution-Aware Methods:** Recent work has focused on designing triggers that better conform to data distributions. Dai et al.[47] extended the semantic backdoor paradigm to GNNs, constructing triggers based on class-relevant node types and graph semantics. Zheng et al.[15] introduced a motif-based framework, where triggers are searched from statistically significant and frequently occurring subgraph structures. They proposed three strategies to enhance effectiveness and stealth: 1) Use of motifs that are absent in the dataset (maximal anomaly), 2) Use of motifs predominantly present in the target class (semantic bias), 3) Use of dense motif structures with high interconnectivity (amplified influence). This taxonomy reflects a growing awareness that

both attack efficacy and stealth must be co-optimized, especially as anomaly detection models are increasingly deployed for backdoor defense.

## C Theoretical Analyses

### C.1 Distributional Shift in Graph vs. Node-Level Attacks

To rigorously compare the detectability of backdoor triggers in graph vs. node classification, we analyze the total variation distance (TV) under local subgraph perturbations.

**Definition 1** (Total Variation Distance). *Let $\mathcal{P}_G, \mathcal{P}'_G$ be the distributions of clean and poisoned graphs in a graph classification task, and $\mathcal{P}_V, \mathcal{P}'_V$ for node classification. The distributional shift is quantified by:*

$$\text{TV}_G := \text{TV}(\mathcal{P}_G, \mathcal{P}'_G), \quad \text{TV}_V := \text{TV}(\mathcal{P}_V, \mathcal{P}'_V), \tag{9}$$

*where TV distance is defined as:*

$$\text{TV}(P, P') := \frac{1}{2} \int_{\mathcal{X}} |p(x) - p'(x)| dx. \tag{10}$$

**Theorem 1** (Lower Bound on Graph-Level Distributional Shift). *Let $N$ be the number of nodes in a graph, $M \ll N$ the number of trigger nodes, and assume the trigger introduces local deviation $\Delta = \|\mu_{trigger} - \mu\|_1$ in node features. Then the total variation distance of the graph-level distribution satisfies:*

$$\text{TV}_G \geq c \cdot \frac{M}{N}, \quad \text{with } c = \frac{1}{2} \cdot \Delta. \tag{11}$$

*Proof.* In graph classification, predictions are made based on aggregated global features (e.g., mean pooling). Let $\mu$ be the average node embedding of a clean graph, and $\mu_{\text{trigger}}$ the embedding from trigger nodes. After injecting a trigger subgraph of size $M$, the poisoned graph has average feature:

$$\mu' = \left(1 - \frac{M}{N}\right) \mu + \frac{M}{N} \mu_{\text{trigger}}.$$

Then the feature shift becomes:

$$\|\mu' - \mu\|_1 = \|\mu_{\text{trigger}} - \mu\|_1 \cdot \frac{M}{N}.$$

Using Pinsker's inequality variant, which states:

$$\text{TV}(P, P') \geq \frac{1}{2} \|\mathbb{E}_{P'}[x] - \mathbb{E}_P[x]\|_1,$$

we obtain:

$$\text{TV}_G \geq \frac{1}{2} \|\mu' - \mu\|_1 = \frac{1}{2} \cdot \|\mu_{\text{trigger}} - \mu\|_1 \cdot \frac{M}{N} = c \cdot \frac{M}{N}.$$

$\square$

**Remark 1** (On Comparison with Node-Level TV). *In node classification, the trigger affects only a few nodes and their k-hop neighbors. Since predictions are localized, the overall distributional shift is often less significant. Letting $\text{TV}_V \approx \delta \cdot \frac{M}{N}$ be a heuristic approximation, we note:*

$$\text{TV}_G \geq c \cdot \frac{M}{N} > \text{TV}_V, \quad \text{if } c > \delta.$$

*This inequality holds when the graph-level aggregation amplifies the statistical impact of the trigger compared to node-local shifts.*

***Caution:*** *The term $\text{TV}_V$ and $\text{TV}_G$ are over different data domains (nodes vs. graphs), and thus are not directly additive:*

$$\text{We cannot claim: } \text{TV}_G \geq \text{TV}_V + c \cdot \frac{M}{N}.$$

*Instead, we highlight the fact that graph-level shift lower bounds node-level shift under same perturbation size, and is strictly greater if the trigger induces global mean shifts.*

**Corollary 1** (Implication for Backdoor Stealth). *Backdoor attacks on graph classification inherently suffer greater distributional shift than on node classification. This makes graph-level triggers more detectable by anomaly detection models, motivating the need for distribution-preserving designs like DPSBA.*

### C.2 Trigger Distributional Detectability Bound

We now establish a lower bound connecting the detectability of backdoor samples to the statistical divergence between clean and poisoned data distributions.

**Definition 2** (Total Variation Distance). *Let $P$ and $P'$ be the distributions over clean and poisoned graphs. Then:*

$$\text{TV}(P, P') := \sup_{A \subseteq \mathcal{G}} |P(A) - P'(A)| = \frac{1}{2} \int_{\mathcal{G}} |p(x) - p'(x)| \, dx, \tag{12}$$

*where $p(x)$, $p'(x)$ are probability densities over $\mathcal{G}$, the space of all graphs.*

**Theorem 2** (Lower Bound on AUC for Detecting Poisoned Graphs). *Let $s(\cdot)$ be any score function used by a binary anomaly detector to distinguish $P$ and $P'$. Then:*

$$\text{AUC}_{det} \geq \frac{1 + \text{TV}(P, P')}{2}. \tag{13}$$

*Proof.* From hypothesis testing theory, the optimal binary decision rule achieves AUC:

$$\text{AUC}_{\text{opt}} = \mathbb{P}(s(x_+) > s(x_-)) = \frac{1 + \text{TV}(P, P')}{2},$$

where $x_+ \sim P'$, $x_- \sim P$. Thus, no detector can achieve lower AUC than this bound. $\qquad \square$

**Corollary 2** (Indistinguishability at Low TV). *If $\text{TV}(P, P') \to 0$, then:*

$$\text{AUC}_{det} \to 0.5,$$

*implying poisoned graphs become statistically indistinguishable from clean ones. Designing triggers that preserve common graph motifs or feature statistics helps satisfy this condition.*

**Implication.** DPSBA minimizes the anomaly loss to reduce $\text{TV}(P, P')$, achieving stealthy attacks even under statistical anomaly detection, as supported by the empirical histograms in Fig. 3.

## D    Algorithm of DPSBA

The DPSBA algorithm is detailed in Algorithm 1. Initially, we train the surrogate model $f_{\theta*}$ using the clean dataset, initialize the relevant model parameters and select poisoning samples from the data whose class is the target class $y_t$ (lines 1-3). From lines 4-8, We inject triggers into the poisoned sample $G_B$. Lines 10-17 and lines 18-25 are topology generation stage and feature generation stage, respectively. The topology (feature) discriminator $\theta_t$ ($\theta_f$) and the generator $\omega_t$ ($\omega_f$) are alternately optimized and then update the surrogate model $f_{\theta*}$ with the current backdoor training set $C'$.

## E    Experimental Details

### E.1    Datasets

The key statistics of the datasets are presented in Table E7.

### E.2    Metrics

This section details the evaluation metrics used to assess both the effectiveness and stealthiness of backdoor attacks, including **Attack Success Rate (ASR)**, **Clean Accuracy Drop (CAD)**, and **Area Under the Curve (AUC)**.

---

**Algorithm 1** DPSBA

---

**Input:** Datasets $C$, target label $y_t$, attack budgets $M$, maximum number of training epochs
**Output:** Backdoor training set $C'$, topology/feature generator parameters $\omega_t/\omega_f$

1: The model $f_{\theta^*}$ is trained through the gradient $\nabla_\theta \mathcal{L}_{train}(C)$;
2: Randomly initialize the topology generator parameters $\omega_t$, the feature generator parameters $\omega_f$, the topology anomaly discriminator parameters $\theta_t$ and the feature anomaly discriminator parameters $\theta_f$;
3: The poisoned samples $G_B$ are selected from $C[y_t]$ according to the confidence score $cfd$ calculated by Eq. (2);
4: **for** $G \in G_B$ **do**
5:   According to the centrality calculated by $\deg(v)/(N-1)$, nodes with high degree centrality are pre-selected as candidates $\mathcal{N}_{can}$;
6:   According to the importance score $S$ calculated by Eq. (3), $M$ injection positions are selected;
7:   Using $m(G; g_t)$ inject triggers in $G$;
8: **end for**
   // Put trigger-embedded graph $G_B$ back into the dataset to form the backdoored dataset $C'$
9: **while** not converged yet **do**
10:   **while** $epoch < epochs$ **do**
11:     Update $\theta_t$ by gradient according to Eq. (5)
12:     Update $\omega_t$ via gradient $\nabla_{\omega_t} \mathcal{L}_{atk}(G_{g_t}(\omega_t)) + \alpha \mathcal{L}_d^{(t)}(D_{\theta_t}(G_{g_t}(\omega_t)))$ according to Eq. (7);
13:   **end while**
14:   **for** $G \in G_B$ **do**
15:     Update $G$ by $m(G; g_t)$;
16:   **end for**
17:   Update $f_{\theta^*}$ by gradient $\nabla_\theta \mathcal{L}_{train}(C')$;
18:   **while** $epoch < epochs$ **do**
19:     Update $\theta_f$ by gradient according to Eq. (6)
20:     Update $\omega_f$ via gradient $\nabla_{\omega_f} \mathcal{L}_{atk}(G_{g_t}(\omega_f)) + \beta \mathcal{L}_d^{(f)}(D_{\theta_f}(G_{g_t}(\omega_f)))$ according to Eq. (8);
21:   **end while**
22:   **for** $G \in G_B$ **do**
23:     Update $G$ by $m(G; g_t)$;
24:   **end for**
25:   Update $f_{\theta^*}$ by gradient $\nabla_\theta \mathcal{L}_{train}(C')$;
26: **end while**
27: **return** $C', \omega_t, \omega_f$

---

Table E7: Datasets statistics

| Datasets | PROTEINS_full | AIDS | FRANKENSTEIN | ENZYMES |
|---|---|---|---|---|
| Number of graphs | 1113 | 2000 | 4337 | 600 |
| Avg. Number of nodes | 39.06 | 15.69 | 16.90 | 32.63 |
| Avg. Number of edges | 72.82 | 16.20 | 17.88 | 62.14 |
| Number of classes | 2 | 2 | 2 | 6 |
| Label distribution | 663 [0], 450 [1] | 400 [0], 1600 [1] | 1936 [0], 2401 [1] | 100 [1-6] |
| Target label | 1 | 0 | 0 | 0 |

**(1) Attack Success Rate (ASR):**   ASR quantifies the proportion of trigger-embedded graphs that are successfully misclassified into the target class. It serves as the primary measure of attack effectiveness and is defined as:

$$ASR = \frac{\text{Number of successful attacks}}{\text{Total number of backdoor trials}}. \tag{14}$$

**(2) Clean Accuracy Drop (CAD):**   CAD measures the performance degradation of the poisoned model on clean samples, indicating the stealthiness of the attack when the trigger is absent. It is defined as the accuracy difference between a clean model and the backdoored model on a clean test

Table E8: Model accuracy on a clean dataset

| Datasets | Model | | |
|---|---|---|---|
| | GCN | GIN | SAGPool |
| PROTEINS_full | 75.58 | 76.23 | 73.99 |
| AIDS | 98.64 | 98.92 | 98.26 |
| FRANKENSTEIN | 60.81 | 64.32 | 62.75 |

set:

$$CAD = ACC_{\text{clean model}} - ACC_{\text{backdoor model}}, \tag{15}$$

where accuracy is calculated as:

$$ACC = \frac{1}{N} \sum_{i=1}^{N} \mathbb{I}(\hat{y}_i = y_i), \tag{16}$$

with $\mathbb{I}(\cdot)$ denoting the indicator function, $\hat{y}_i$ the predicted label, $y_i$ the ground-truth label, and $N$ the number of test samples.

**Interpretation:** A high ASR indicates effective attack performance, while a low CAD implies minimal disruption to normal prediction behavior, reflecting strong stealth.

**(3) Area Under the Curve (AUC):** To evaluate stealth from the perspective of anomaly detection, we use the AUC score produced by an anomaly detection model (e.g., SIGNET) trained on clean graphs. AUC reflects the model's ability to distinguish backdoor samples from clean ones:

$$AUC = P(s(x_+) > s(x_-)), \tag{17}$$

where $s(x)$ denotes the anomaly score of graph $x$, and $x_+, x_-$ are randomly selected anomalous and clean samples, respectively.

**Interpretation:**

- AUC $\to$ 1.0: the detector perfectly separates anomalies from clean data, i.e., low stealth.
- AUC $\to$ 0.5: the detector cannot distinguish between backdoor and clean samples,i.e., high stealth.
- AUC $<$ 0.5: the detector is misled, incorrectly prioritizing clean samples as more anomalous.

**Goal:** An effective backdoor should aim for an AUC as close as possible to 0.5, indicating that the anomaly introduced by the trigger is indistinguishable from natural variations in clean data, thus enhancing stealth against detection models.

### E.3   Accuracy of Graph Classifiers on Clean Data

Table E8 illustrates the accuracy of three different graph classifiers on the clean datasets.

### E.4   Impact of the Poisoning Rate

To investigate how the poisoning rate influences DPSBA's performance, we evaluate the variation of attack success rate (ASR) and clean accuracy drop (CAD) under different poisoning rates. Since the anomaly level of the trigger is determined by its structural and feature design rather than injection frequency, we focus here on ASR and CAD as shown in Figure E6. As the poisoning rate increases from 1% to 7%, ASR consistently improves across all datasets, confirming that injecting more poisoned samples strengthens the backdoor effect. However, the growth rate of ASR gradually slows, indicating diminishing returns. Meanwhile, CAD shows a slight upward trend with higher poisoning rates, but remains below 5% in all cases, demonstrating that DPSBA maintains high stealth even under increased poisoning levels. This result suggests that while increasing the poisoning rate enhances attack effectiveness, DPSBA can still preserve stealthiness due to its clean-label and anomaly-aware design. Practitioners may adjust the poisoning budget to balance performance and resource cost without significantly sacrificing stealth.

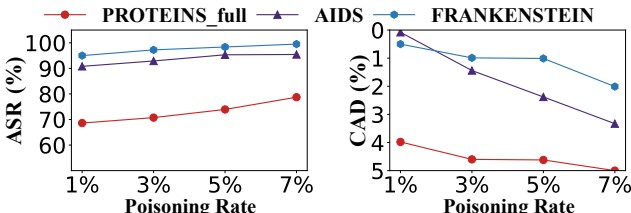

Figure E6: The impact of the Poisoning Rate

## E.5 Impact of Trigger Size

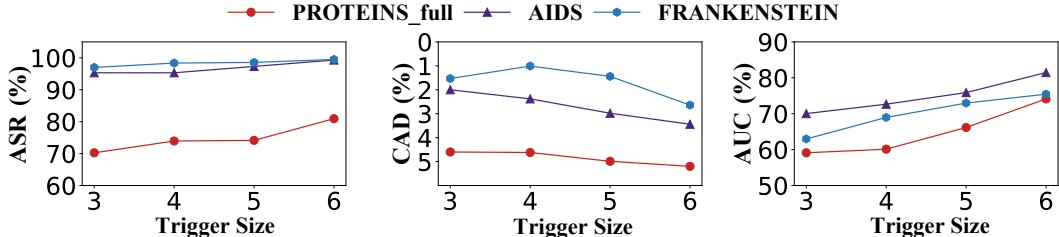

Figure E7: The impact of trigger size

We further examine the effect of trigger size on the performance of DPSBA. As shown in Figure E7, increasing the size of the trigger leads to a consistent improvement in attack success rate (ASR) across all datasets. This is expected, as larger triggers introduce stronger perturbations to the graph's topology and features, making it easier for the model to associate the injected pattern with the target label. However, this enhancement in effectiveness comes at the cost of stealthiness. As the size of the trigger increases, the level of anomaly, as measured by the AUC, also increases. This indicates that larger triggers deviate more from the normal graph distribution, thus being more detectable by anomaly detection models. Still, the AUC values remain within an acceptable range (mostly below 80%), preserving a reasonable degree of stealth. Moreover, the clean accuracy drop (CAD) exhibits a general upward trend with increasing trigger size, though it remains under 5% in most cases. This suggests that even larger triggers do not significantly impair the model's generalization performance on clean data. In summary, trigger size presents a clear trade-off: larger triggers yield stronger attacks but increase detectability, while smaller ones offer greater stealth at the cost of ASR. DPSBA supports flexible adjustment of this trade-off based on practical requirements.

## E.6 Higher ASR in Cross-Architecture Transfer Settings

We explain this counterintuitive phenomenon from three perspectives: **model expressiveness**, **trigger generalizability**, and **dataset characteristics**.

**(1) Model Expressiveness vs. Overfitting:** GIN and SAGPool are more expressive than GCN. GIN is theoretically equivalent to the Weisfeiler-Lehman test, capable of distinguishing fine-grained substructures and SAGPool uses self-attention to highlight global structures most relevant for classification. These powerful models, when used as surrogates, tend to learn highly specialized, structure-sensitive triggers. While effective on the surrogate model itself, such triggers are prone to overfitting and may fail to generalize to unseen samples or to minor parameter shifts. This aligns with prior observations [48] that stronger surrogates often overfit their own gradients, reducing perturbation transferability.

**(2) GCN-Trained Triggers Are More Transferable:** GCN, with its smoother message passing and lower expressiveness, tends to learn broader and more transferable trigger patterns. These triggers may achieve lower ASR on GCN itself (as seen in Table 1), but generalize better when transferred to expressive models like GIN or SAGPool, which are more sensitive to subtle perturbations and thus amplify the attack effect.

**(3) Dataset-Specific Factors Amplify the Effect:** In the AIDS dataset, molecular graphs are small but structurally complex. Local substructures (e.g., functional groups) are crucial, making GIN highly sensitive to local changes caused by transferred triggers. In PROTEINS_full dataset, graphs are larger and denser (node–edge ratio $\approx$ 1:1.86), favoring models like SAGPool that focus on global structure. GCN's triggers are naturally aligned with such global perturbations, which SAGPool tends to emphasize.

# F   Time Complexity Analysis

We analyze the training time complexity of DPSBA by accounting for graph sparsity and the dimensionality of node features. The framework consists of three main stages: (1) hard sample selection; (2) trigger location selection and initialization; and (3) adversarial optimization of the topology and feature generators. Let us denote:

- $C$: number of graphs in the clean training set
- $N$: average number of nodes per graph
- $d$: average degree of nodes (assumes sparse graph, i.e., $d \ll N$)
- $F$: node feature dimension
- $B$: the total number of poisoned graphs
- $E$: total number of adversarial training epochs

**1) Hard Sample Selection.** For each graph in the target class, we compute its confidence score with respect to the target label using a forward pass through the surrogate model. Assuming each node aggregates information from $d$ neighbors, and node features are $F$-dimensional, the per-graph complexity is $\mathcal{O}(N \cdot d \cdot F)$, leading to $\mathcal{O}(C \cdot N \cdot d \cdot F)$. This term reflects the initial overhead of scoring and ranking all graphs to identify low-confidence (hard) samples.

**2) Trigger Location Selection.** For each selected hard sample, DPSBA computes node importance via a deletion-based interpretability scheme. Each candidate node requires one additional forward pass through the GNN after node removal. If we sample $k$ candidates per graph, the cost becomes $\mathcal{O}(k \cdot N \cdot d \cdot F)$. Since $k$ is fixed and small ($k \approx 2M$), this step scales as $\mathcal{O}(C \cdot N \cdot d \cdot F)$. Thus, steps (1) and (2) share the same leading complexity.

**3) Trigger Optimization.** In each epoch, we optimize the topology and feature generators to simultaneously (a) maximize attack loss and (b) minimize anomaly detection confidence through adversarial discriminators (GCN and MLP). Each poisoned graph in the batch undergoes GCN-based classification and gradient updates. Assuming adversarial training lasts $E$ epochs, the total cost becomes $\mathcal{O}(E \cdot B \cdot N \cdot d \cdot F)$. This dominates the runtime due to repeated gradient steps.

**Overall Complexity.** Combining the above three stages, the total training complexity of DPSBA is: $\mathcal{O}(C \cdot N \cdot d \cdot F + E \cdot B \cdot N \cdot d \cdot F)$ This expression reveals linear scalability with respect to the number of nodes, average degree, and feature dimension. The first term corresponds to one-time sampling and location inference; the second reflects iterative adversarial training. As the number of poisoned graphs ($B$) and training epochs ($E$) are relatively small, DPSBA maintains practical efficiency under sparse graph assumptions. DPSBA achieves a favorable balance between attack efficacy and efficiency: its overall complexity is comparable to standard GNN training pipelines while introducing minimal overhead. Moreover, by leveraging sparse message passing and modular optimization, DPSBA remains scalable to real-world graphs with thousands of nodes and high-dimensional features.

In addition, we benchmark the actual training time of DPSBA and compare it with several representative baselines (ER-B, LIA, GTA, and Motif) on the largest dataset, **FRANKENSTEIN**. The results are reported in Table F9. All experiments are conducted on a machine equipped with a 14-core Intel i7-12700H CPU, an NVIDIA GeForce RTX 3060 GPU (12 GB), and Windows 11 (version 23H2). As shown in the table, DPSBA achieves the highest ASR (99.84%) while maintaining the lowest CAD (1.83%) and AUC (73.46%), demonstrating a superior balance between attack success and stealth. Although DPSBA incurs slightly more training time compared to the lightest baseline (ER-B), the overhead is acceptable given the substantial gains in performance. Notably, our framework integrates an *early-stopping mechanism* during adversarial training, which adaptively terminates optimization once the attack objective converges. This not only reduces computational overhead but also avoids unnecessary overfitting, making DPSBA both effective and time-efficient.

Table F9: Training Time and Attack Performance on FRANKENSTEIN.

| Metrics | ER-B | LIA | GTA | Motif | Ours |
|---|---|---|---|---|---|
| ASR (%) | 92.06 | 82.63 | 95.23 | 84.56 | 99.84 |
| CAD (%) | 3.60 | 2.35 | 1.95 | 2.44 | 1.83 |
| AUC (%) | 85.73 | 76.15 | 91.06 | 87.54 | 73.46 |
| Time (s) | 210 | 261 | 376 | 239 | 281 |

# G    Discussion of Inductive Graph Classification Setting

While our current work focuses on the transductive graph classification setting, we believe that several key components of DPSBA have promising potential to extend to the inductive regime. In particular: 1) The feature generator operates based on localized structural and attribute information at the trigger injection site, and 2) The distribution-aware discriminators regularize stealth on a per-graph basis through adversarial training. Both modules are graph-local in nature and do not rely on inter-graph interactions or train–test graph overlap, making them amenable to inductive settings where new test graphs are unseen during training. That said, certain components, such as the hard sample selection module, currently assume access to sufficient examples of the target class, which poses challenges in few-shot scenarios. Adapting this component to work under strong data constraints (e.g., via meta-learning or class-agnostic proxy supervision) is a meaningful direction for future work.

# H    Broader Impacts

This work reveals that even under clean-label settings, graph classification models remain highly vulnerable to stealthy backdoor attacks. By exposing this underexplored threat, our method highlights the limitations of existing defenses and underscores the need for more robust anomaly detection and training strategies. While our approach may be misused, we release it to raise awareness and promote the development of secure graph learning systems in critical domains such as bioinformatics, finance, and cybersecurity.

