# OpenReview forum: "Stealthy Yet Effective: Distribution-Preserving Backdoor Attacks on Graph Classification"
_NeurIPS.cc/2025/Conference — NeurIPS 2025 poster_

### Official Review · Reviewer_PXmz · 2025-06-28

**Clarity:** 3
**Significance:** 3
**Originality:** 3
**Rating:** 5
**Confidence:** 3

**Summary:**

The authors proposed a dynamic trigger generation method for data poisoning in the graph classification task. Existing methods mostly inject a fixed subgraph into source class graphs. This makes them prone to be detected by anomaly based detectors such as SIGNET. The authors made several changes: 1) Inject triggers only into the target class graphs. 2) Use optimisation to dynamically generate subgraphs and features adaptive to the original graph. They empirically show that in three datasets, the new method can achieve better ASR while lowering the chance of being detected.

**Questions:**

In Table 2, why did GIN+AIDS and SAGPool+PROTEINS  achieve better ASR compared with Table 1?

**Ethical Concerns:**

["NO or VERY MINOR ethics concerns only"]

**Final Justification:**

Other reviews and the rebuttal confirm my assessment that the paper is of good quality.

**Paper Formatting Concerns:**

Please use mathrm for cfd and diff.

**Quality:**

3

**Strengths And Weaknesses:**

Strengths:
- The proposed method is well motivated, and the components are well justified.
- The writing is clear in most places.

Weaknesses
- The performance seems to be sensitive to the surrogate model. It is unclear how robust surrogate model choice and training are to affect the reported results.
- Adaptive trigger can be sensitive. There is a lack of theoretical guarantee that it works across different graph neural networks.

---

> ### Author Rebuttal · Authors · 2025-07-28
>
> ## **Response to Comments on Surrogate Model Robustness (Weaknesses 1)**
>
> We sincerely thank the reviewer for pointing out the importance of surrogate model robustness. This is indeed a critical concern for the practical viability of backdoor attacks in real-world settings.
>
> In most realistic attack scenarios, the attacker **does not have access to the architecture or weights of the actual deployed model** (i.e., the "victim model"). As a result, backdoor attack frameworks must rely on surrogate models that are both **expressive** and **transferable**, enabling triggers to generalize across unknown target architectures. In our work, we adopt **GCN, GIN, and SAGPool** as representative surrogates precisely because they vary in architectural design and have been widely adopted in prior works such as **Motif**, **Motif-S**, and **ER-B**, allowing for meaningful and consistent comparisons.
>
> It is expected that the **Attack Success Rate (ASR)** may vary depending on the surrogate-target pair, as different target GNNs have inherently different robustness and inductive biases. However, our proposed **DPSBA consistently achieves high ASR and low CAD across all surrogate models**, demonstrating strong generalization across architectures. Additionally, our use of **distribution-aware discriminators** promotes stealthiness in a model-agnostic manner, mitigating overfitting to any single surrogate.
>
> We agree that evaluating additional architectures would further validate robustness, and we plan to explore this in future work.
>
> ---
>
> ## **Response to Comments on Adaptive Trigger Robustness and Theoretical Guarantees (Weaknesses 2)**
>
> We appreciate the reviewer’s comment regarding the robustness of adaptive triggers and the need for theoretical guarantees. This is indeed a crucial issue for ensuring transferability across different GNNs.
>
> To this end, our work presents a formal theoretical analysis in **Appendix C.2**, where we derive the **Trigger Distributional Detectability Bound**. This result establishes a lower bound on anomaly detection AUC based on the **total variation distance (TVD)** between clean and poisoned graph distributions. Importantly, this analysis is model-agnostic, depending only on distributional divergence rather than any specific GNN internals. Thus, it offers theoretical support for the stealth–effectiveness trade-off of adaptive triggers.
>
> Empirically, we evaluate DPSBA across diverse GNNs (GCN, GIN, SAGPool) spanning a range of inductive biases. DPSBA maintains high ASR and stealth performance in all cases, confirming the robustness of the adaptive trigger mechanism. We appreciate the reviewer’s suggestion and will consider extending our theoretical analysis to additional architectures in the future.
>
> ---
>
> ## **Response to Comments on Higher ASR in Cross-Architecture Transfer Settings (Question)**
>
> We thank the reviewer for highlighting the interesting observation in Table 2, where **GCN→GIN+AIDS** and **GCN→SAGPool+PROTEINS** achieve higher ASR than their white-box counterparts (GIN→GIN+AIDS, SAGPool→SAGPool+PROTEINS). Though initially counterintuitive, this phenomenon can be explained from three perspectives: **model expressiveness**, **trigger generalizability**, and **dataset characteristics**.
>
> **(1) Model Expressiveness vs. Overfitting**
> GIN and SAGPool are more expressive than GCN:
> - *GIN* is theoretically equivalent to the Weisfeiler-Lehman test, capable of distinguishing fine-grained substructures.
> - *SAGPool* uses self-attention to highlight global structures most relevant for classification.
>
> These powerful models, when used as surrogates, tend to learn highly specialized, structure-sensitive triggers. While effective on the surrogate model itself, such triggers are prone to **overfitting** and may fail to generalize to unseen samples or to minor parameter shifts. This aligns with prior observations (e.g., *Tramèr et al., ICLR 2018*) that stronger surrogates often overfit their own gradients, reducing perturbation transferability.
>
> > Tramèr, F. et al. (2018). Ensemble adversarial training: Attacks and defenses. ICLR.
>
> **(2) GCN-Trained Triggers Are More Transferable**
> GCN, with its smoother message passing and lower expressiveness, tends to learn broader and more transferable trigger patterns. These triggers may achieve lower ASR on GCN itself (as seen in Table 1), but generalize better when transferred to expressive models like GIN or SAGPool, which are more sensitive to subtle perturbations and thus **amplify the attack effect**.
>
> **(3) Dataset-Specific Factors Amplify the Effect**
> - *AIDS*: Molecular graphs are small but structurally complex. Local substructures (e.g., functional groups) are crucial, making GIN highly sensitive to local changes caused by transferred triggers.
> - *PROTEINS_full*: Graphs are larger and denser (node–edge ratio ≈ 1:1.86), favoring models like SAGPool that focus on global structure. GCN’s triggers are naturally aligned with such global perturbations, which SAGPool tends to emphasize.
>
> **In summary:**
> - Strong surrogates (GIN/SAGPool) may overfit to narrow trigger distributions, limiting transferability;
> - GCN, though less optimal in white-box settings, produces **robust and transferable triggers**;
> - These results demonstrate DPSBA's strong generalization under model mismatch and strengthen its practical utility.
>
> We will incorporate this discussion in the revised version of our paper.
>
> ---
>
> ## **Response to Paper Formatting Concerns**
>
> We thank the reviewer for pointing out the formatting issue. We will revise the paper to use `\mathrm{}` consistently for all mathematical expressions involving operators such as `cfd` and `diff`.

---

> > ### Author Response · Authors · 2025-08-06
> >
> > We sincerely thank the reviewer for the positive evaluation and support of our work. If there are any remaining concerns or suggestions, we would be very happy to discuss and clarify them.

---

> > > ### Comment · Reviewer_PXmz · 2025-08-07
> > > **Thanks for the rebuttal**
> > >
> > > I am happy with the response.

---

> > > > ### Author Response · Authors · 2025-08-08
> > > >
> > > > Thank you for your positive feedback. We appreciate your recognition of our work.

---

### Official Review · Reviewer_oU5D · 2025-06-30

**Clarity:** 3
**Significance:** 3
**Originality:** 3
**Rating:** 5
**Confidence:** 4

**Summary:**

This paper introduces DPSBA, a clean-label backdoor attack framework for graph classification that uniquely preserves both structural and semantic distributional properties of clean graphs. Unlike existing methods that rely on structurally anomalous or semantically conflicting triggers, DPSBA generates in-distribution subgraph triggers using adversarial training with anomaly-aware discriminators. The paper demonstrates strong results in terms of attack success rate (ASR), stealth (AUC), and transferability across architectures, while maintaining negligible performance drop on clean samples (CAD). The methodology is well-motivated, technically sound, and supported by extensive empirical and theoretical validation.

**Questions:**

1.	Could you discuss how DPSBA would behave in multi-class settings, especially when there is no clear minority target class?

2.	Can DPSBA be adapted for inductive settings (e.g., few-shot graph classification)?

**Ethical Concerns:**

["NO or VERY MINOR ethics concerns only"]

**Final Justification:**

I appreciate the authors' detailed response. The clarifications provided have largely resolved my concerns. I will retain my original score. Thank you for your work

**Limitations:**

yes

**Quality:**

3

**Strengths And Weaknesses:**

Strengths:

1.	The paper identifies and formalizes the dual-source anomaly problem (structural + semantic) in graph-level backdoor attacks and convincingly shows why existing methods fail in stealth.

2.	The paper provides clear theoretical bounds on the total variation distance and its relationship to anomaly detectability (AUC), which is rarely discussed in prior graph backdoor literature.

3.	The clean-label, limited-poisoning setting significantly increases the practical relevance and threat realism of the proposed method.

Weaknesses:

1.	Although trigger size is fixed to 4, it would be interesting to show how ASR and AUC vary with trigger size.

2.	All datasets used are binary. It’s unclear whether DPSBA’s strategy generalizes to multi-class graph classification tasks.

3.	The overall writing requires more consistency and standardization in expression.

---

> ### Author Rebuttal · Authors · 2025-07-29
>
> ## **Response to Comments on Trigger Size (Weaknesses 1)**
>
> We sincerely thank the reviewer for pointing out the impact of trigger size changes on the performance of DPSBA.
>
> In Appendix E.6, we provide a detailed analysis of how trigger size variations impact the three evaluation metrics of DPSBA, with experimental results illustrated in Figure E7. The findings reveal a distinct trade-off relationship regarding trigger size: larger triggers yield stronger attacks but increase detectability, while smaller ones offer greater stealth at the cost of ASR. DPSBA supports flexible adjustment of this trade-off based on practical requirements.
>
> ---
>
> ## **Response to Comments on Multi-class Graph  Classification Task Experiment (Weaknesses 2 & Question 1)**
>
> We sincerely thank the reviewer for raising the important concern about the performance of our approach in multi-class graph classification tasks.
>
> Following the reviewer’s suggestion, we additionally evaluated our method on the **ENZYMES** dataset (a 6-class biomolecular classification task), which is finer-grained and more reflective of real-world scenarios. As shown below, **DPSBA consistently outperforms** all baselines in terms of both attack success rate (ASR) and stealth (CAD, AUC) across three surrogate models. Notably, most baselines (except GTA) fail on this dataset due to high attribute and strucutre variability, underscoring the advantage of our distribution-preserving and adaptive design.
>
> | Dataset | Surrogate Model | Metric | ER-B  | LIA   | GTA   | Motif | Motif-S   | Ours      |
> | ------- | --------------- | ------ | ----- | ----- | ----- | ----- | --------- | --------- |
> | ENZYMES | GCN             | ASR    | 26.09 | 30.43 | 95.33 | 21.74 | 15.21     | **96.67** |
> |         |                 | CAD    | 4.17  | 4.99  | 3.00  | 4.99  | **-1.67** | -0.67     |
> |         |                 | AUC    | 68.32 | 66.15 | 71.20 | 71.35 | 66.22     | **66.11** |
> |         | GIN             | ASR    | 37.83 | 27.02 | 96.00 | 16.21 | 12.16     | **99.33** |
> |         |                 | CAD    | 9.17  | 10.00 | 2.67  | 8.33  | 4.17      | **-0.33** |
> |         |                 | AUC    | 71.40 | 62.01 | 76.42 | 68.18 | 65.78     | **41.20** |
> |         | SAGPool         | ASR    | 29.54 | 38.63 | 100.0 | 15.91 | 11.37     | **100.0** |
> |         |                 | CAD    | 4.33  | 6.67  | 5.00  | 10.83 | **3.33**  | 4.00      |
> |         |                 | AUC    | 57.73 | 63.98 | 70.37 | 75.47 | 69.48     | **49.41** |
>
> We will include these results and the corresponding analysis in the revised paper.
>
> ---
>
> ## **Response to Comments on Writing (Weaknesses 3)**
>
> We thank the reviewer for pointing out the consistency and standardization of writing. We will focus on the consistency and standardization of writing in the revised paper.
>
> ---
>
> ## **Response to Comments on Inductive Settings (Question 2)**
>
> We sincerely thank the reviewer for raising this insightful and forward-looking question regarding the potential adaptation of our framework to inductive scenarios, such as few-shot graph classification.
>
> While our current work focuses on the transductive graph classification setting, we believe that several key components of DPSBA have promising potential to extend to the inductive regime. In particular:
>
> - The **feature generator** operates based on localized structural and attribute information at the trigger injection site, and
> - The **distribution-aware discriminators** regularize stealth on a per-graph basis through adversarial training,
>
> Both modules are **graph-local** in nature and do not rely on inter-graph interactions or train–test graph overlap, making them amenable to inductive settings where new test graphs are unseen during training.
>
> That said, certain components, such as the hard sample selection module, currently assume access to sufficient examples of the target class, which poses challenges in few-shot scenarios. Adapting this component to work under strong data constraints (e.g., via meta-learning or class-agnostic proxy supervision) is a meaningful direction for future work.
>
> We will include this discussion and future perspective in the revised version of the paper. We again thank the reviewer for this valuable suggestion.

---

> > ### Comment · Reviewer_oU5D · 2025-08-06
> >
> > I appreciate the authors' detailed response. The clarifications provided have largely resolved my concerns. I will retain my original score. Thank you for your work.

---

> > > ### Author Response · Authors · 2025-08-06
> > >
> > > Thank you for your time and thoughtful comments. We appreciate your acknowledgment of our clarifications, and we’re grateful for your constructive feedback throughout the review process.

---

### Official Review · Reviewer_5SVM · 2025-06-30

**Clarity:** 3
**Significance:** 3
**Originality:** 3
**Rating:** 5
**Confidence:** 3

**Summary:**

This paper proposes DPSBA, a clean-label backdoor attack for graph classification tasks. It introduces a strategy to learn distribution-preserving triggers that reduce both structural and semantic anomalies, which are commonly exploited by anomaly detectors. The method leverages a two-stage adversarial training framework involving a surrogate classifier and two discriminators that penalize detectability. The approach is evaluated on three binary classification datasets, showing strong attack success with low anomaly scores (AUC), outperforming prior baselines in stealth and effectiveness.

**Questions:**

1. Could the authors clarify how the feature generator avoids distribution shift when the target graphs have high attribute variance?
2. How was the surrogate model selected and does its architecture affect DPSBA performance?

**Ethical Concerns:**

["NO or VERY MINOR ethics concerns only"]

**Final Justification:**

The author has effectively and thoroughly addressed all of my concerns during the rebuttal process. As a result, I am willing to raise my score.

**Quality:**

3

**Strengths And Weaknesses:**

Strengths:
1. Graph-level backdoor attacks are relatively under-studied compared to node-level attacks. This paper contributes meaningfully by focusing on stealth in the full-graph setting, which is more difficult due to global representation constraints.
2. The clean-label assumption improves realism and attack stealth, and makes the threat model more aligned with practical concerns in sensitive domains like bioinformatics or chemical compound classification.
3. The paper presents a number of informative ablation studies that dissect the contributions of different modules in DPSBA, and the visualization of anomaly score distributions adds credibility to stealth claims.

Weaknesses
1. The training pipeline involves adversarial optimization over both structure and feature discriminators, requiring repeated updates to a surrogate classifier. While theoretically motivated, this complexity could hinder adoption, especially on larger datasets.
2. The role of hyperparameters α and β in balancing anomaly suppression vs. attack success could be elaborated further, particularly in the main paper (rather than only in the appendix).
3. It would be informative to include a brief discussion of scenarios where DPSBA fails (e.g., highly homophilic graphs or extremely sparse graphs), to guide practitioners.
4. The clean-label setting is well motivated, but the paper could benefit from a more direct comparison (qualitative or conceptual) with label-flipping approaches to highlight its practical trade-offs.

---

> ### Author Rebuttal · Authors · 2025-07-29
>
> ## **Response to Comment on Complexity (Weaknesses 1)**
>
> We appreciate the reviewer’s insightful suggestion regarding the complexity of DPSBA.
>
> In Appendix F, we conducted a detailed analysis of the time complexity from three aspects: Hard Sample Selection, Trigger Location Selection, and Trigger Optimization. Furthermore, we compared the execution time of DPSBA with other baselines under the same conditions using the largest dataset, FRANKENSTEIN. The experimental results are presented in Table F9. The results demonstrate that DPSBA is slightly higher than but remains comparable to other baselines. Moreover, DPSBA outperforms the other baselines in terms of both attack effectiveness and stealth.
>
> ---
>
> ## **Response to Comments on  Elaborating Hyperparameters $α$ and $β$ (Weaknesses 2)**
>
> We sincerely thank the reviewer for raising the important concern about elaborating the role of hyperparameters $α$ and $β$ in balancing anomaly suppression vs. attack success. In **Appendix E.5**, we provide a detailed analysis of the hyperparameters $α$ and $β$, with Figure E5 illustrating their impact on ASR and AUC. Subsequently, we will incorporate your valuable suggestions and revise the paper to elaborate on these impacts in the main text.
>
> ---
>
> ## **Response to Comments on the Failure Situation of DPSBA (Weaknesses 3)**
>
> We sincerely appreciate the reviewers' concern regarding the failure of DPSBA. Below, we will briefly discuss the scenarios that may lead to the failure of DPSBA.
>
> **1. Dense graphs**
>
> The **PROTEINS_full** dataset exhibits a relatively high graph density, yet its Attack Success Rate (ASR) is significantly lower than those of the other two datasets. We thus hypothesize that **graph density may influence the attack success rate of DPSBA**. This phenomenon can be attributed to the fact that in dense graphs, node connections are already highly abundant, making trigger injection less impactful on the overall graph structure. Moreover, since nodes in dense graphs typically have numerous neighbors, it becomes more challenging for an attacker to significantly alter global or local graph patterns (e.g., degree distribution, substructure) through minor topological modifications.
> In contrast, sparse graphs contain fewer edges, meaning each edge plays a more critical role in determining structural properties. Consequently, attackers can more effectively manipulate key paths or neighborhood relationships using simpler trigger structures, thereby achieving higher attack efficacy.
>
> **2. Highly homophilic graphs**
>
> In graph classification backdoor attacks, achieving a high attack success rate (ASR) requires significant modifications to local structures (e.g., by inserting trigger subgraphs), which inevitably induces a noticeable shift in the graph's homophily distribution. Conversely, ensuring high attack stealth demands minimal deviation in the homophily distribution of poisoned graphs. However, this constraint inherently limits the trigger's ability to effectively alter model predictions, resulting in reduced ASR.
> In highly homophilic graphs, trigger injection may initially appear effective for successful attacks. Yet, such triggers often introduce obvious deviations in homophily distribution compared to the original graphs, making them easily detectable by defense mechanisms and thus severely compromising stealth. DPSBA aims to strike a balance between ASR and stealth. However, in highly homophilic graphs, the stringent constraints create an optimization dilemma: the adversarial optimization process struggles to concurrently satisfy both objectives, frequently leading to attack failure.
>
> ---
>
> ## **Response to Comment on Clean-label Setting (Weaknesses 4)**
>
> We appreciate the reviewer’s insightful suggestion regarding the clean-label setting.
>
> In Appendix A, we elaborate on the role of the clean-label setting. A comparative analysis of the experimental results in Tables A5 and A6 reveals that **without the clean-label setting**, all baseline methods achieve AUC values exceeding 89%, indicating their attacks exhibit poor stealth. By contrast, **under the clean-label setting**, although the Attack Success Rate (ASR) of all methods decreases, the AUC metric also declines correspondingly. This demonstrates that attacks conducted in the clean-label setting possess superior stealth compared to their non-clean-label counterparts.
>
> ---
>
> ## **Response to Comment on Avoiding Distribution Shift (Question 1)**
> We thank the reviewer for raising this important point.
>
> In our framework, the feature generator does  not rely on distribution priors of the entire dataset. Instead, it takes as input the local structural and attribute context around the trigger injection site and generates features that blend smoothly with the surrounding neighborhood.
> This allows the generator to adapt to the specific variance of each target graph on a per-instance basis, without requiring any handcrafted normalization or explicit regularization. As a result, the generated features remain close to the local data manifold, reducing the likelihood of creating detectable anomalies, even in the presence of high attribute variance across graphs.
> We will revise the paper to make this aspect of the generator design clearer in the main text.
>
> ---
>
> ## **Response to Comment on Surrogate Model  (Question 2)**
>
> We thank the reviewer for the thoughtful question.
>
> In most realistic attack scenarios, the attacker does **not** have access to the architecture or parameters of the actual deployed model (i.e., the "victim model"). Therefore, backdoor attack frameworks must rely on **surrogate models** during training. These surrogate models need to be both **expressive** (to enable learning effective triggers) and **transferable** (so that the learned triggers can generalize to unknown architectures).
>
> In our work, we select **GCN**, **GIN**, and **SAGPool** as surrogate models based on two main considerations:
>
> 1. **Architectural Diversity**: These models vary significantly in message-passing mechanisms and inductive biases—GCN performs neighborhood averaging, GIN emphasizes substructure discrimination, and SAGPool captures global context via self-attention pooling. This diversity allows us to test the generalization ability of our method across representative GNN families.
>
> 2. **Relevance to Prior Work**: All three models are widely adopted in recent graph backdoor baselines (e.g., Motif, Motif-S, ER-B), providing a fair and widely accepted evaluation protocol.
>
> As for performance sensitivity:
> Yes, the **architecture of the surrogate model can affect the attack success rate (ASR)** due to different robustness properties of target models. However, our proposed **DPSBA consistently achieves high ASR and low CAD across all surrogates**, indicating strong **robustness to surrogate choice**.
>
> We agree that broader evaluation on additional models such as GAT or GraphSAGE would offer further insights, and we plan to include such extensions in future work.

---

### Official Review · Reviewer_ePuR · 2025-07-02

**Clarity:** 3
**Significance:** 2
**Originality:** 2
**Rating:** 4
**Confidence:** 5

**Summary:**

This paper introduces DPSBA, a novel clean-label backdoor framework. DPSBA uniquely employs adversarial training, guided by anomaly-aware discriminators, to learn in-distribution triggers. By effectively suppressing both structural and semantic anomalies, DPSBA achieves high attack success rates while significantly enhancing stealth. Extensive experiments on real-world datasets demonstrate that DPSBA strikes a superior balance between effectiveness and detectability compared to existing state-of-the-art baselines.

**Questions:**

1.	Universal Trigger： The method focuses on generating a trigger for each individual sample. Have you explored the possibility of developing a universal trigger that is transferable across different samples? Demonstrating the feasibility or discussing the challenges of a more transferable trigger would elevate the contribution and applicability of your work.

2.	Problem Hardness and Design Rationale：Please articulate why this problem is challenging and the specific insights guiding your method design choices. This would justify your approach and strengthen the originality.

3.	Comprehensive Experimental Analysis：There is only the result of a 5% poisoning rate. What are the results if this rate is lowered? For the PROTEINS_full dataset, the Attack Success Rate (ASR) ranges from 73% to 94%. Can you investigate at what poisoning rate the ASR on PROTEINS_full approaches the 98%-99% achieved on the AIDS or FRANKENSTEIN datasets?

Evaluations on more benchmark datasets.

Discuss the possibility of generalizing the proposed attack on other graph-relevant tasks

**Ethical Concerns:**

["NO or VERY MINOR ethics concerns only"]

**Final Justification:**

The response has addressed my questions, and I raise my score from 3 to 4 accordingly.

**Limitations:**

While the authors discuss the methodological limitations of their work, they do not address its potential negative societal impact.

**Quality:**

3

**Strengths And Weaknesses:**

Strengths:

+Paper structure: The paper is well-structured. It first demonstrates how existing methods suffer from distribution shifts, which motivates the proposed approach and experimental validation. This makes the paper easy to follow and understand.

+Writing: The writing is clear and concise.

Weaknesses

-Methodology：The proposed method generates a trigger for each individual sample. There is no discussion about the transferability of these sample-specific triggers across different samples or the potential for developing a universal trigger. Addressing these would enhance the practical utility and its contribution.

-Originality：The paper leverages existing methods to achieve positive results for the task, the novelty appears limited. To strengthen its originality, the authors could highlight specific insights or unique theoretical analysis.

-Experiment：
The experiments only consider graph classification tasks and three datasets. Is this possible to generalize to other graph-relevant tasks and more benchmark datasets, including real-world ones.

The evaluation does not cover the impact of crucial parameters such as poisoning rate and trigger magnitude on the attack performance.

---

> ### Author Rebuttal · Authors · 2025-07-28
>
> ## **Response to Comments on Trigger Transferability and Universal Trigger Design (Weaknesses-Methodology & Question 1)**
>
> We thank the reviewer for raising this insightful question regarding the feasibility of **universal triggers** and the transferability of our sample-specific design.
>
> But currently, the existing graph backdoor methods, both generative  (e.g., GTA) and search-based (e.g., Motif), adopt **sample-specific triggers**. This is primarily due to the **heterogeneous nature of real-world graph data**, where node attributes and topological structures vary significantly across samples. A trigger that appears inconspicuous in one graph may be highly anomalous in another.
>
> Our method follows this paradigm but further **incorporates anomaly-aware regularization** to explicitly enforce stealth during trigger generation. In contrast, prior sample-specific methods like GTA focus primarily on effectiveness and do not explicitly address detectability.
>
> We agree that universal triggers could improve training efficiency and generalization, and are highly desirable in practical deployments. However, designing such triggers poses substantial challenges:
>
> - **Structural & Distributional Variability**: In graph-level classification, the diversity across samples makes it difficult to craft a fixed trigger that maintains both semantic relevance and distributional alignment across the dataset.
> - **Stealth vs. Effectiveness Trade-off**: As shown in Table 1,  fixed triggers such as **Motif** achieve high ASR (e.g., 92.69% on AIDS under GCN), but suffer from extremely high anomaly scores (AUC = 99.71%). When stealth is improved by using more frequent motifs (Motif-S), the ASR drops significantly (to 56.08%). This highlights the inherent trade-off between universality and stealth.
>
> We appreciate the reviewer’s suggestion and agree that developing **universal or class-conditional trigger generator** that balance **generalizability and stealth** is a promising and underexplored research direction.  We will add this discussion to the revised version of our paper and are enthusiastic to pursue this direction in future work. We thank the reviewer again for this valuable feedback.
>
> ---
>
> ## **Response to Comments on Originality and Problem Hardness (Weakness – Originality & Question 2)**
>
> We thank the reviewer for highlighting the importance of originality.
>
> Graph-level backdoor attacks are far less studied, especially under **clean-label** and **distribution-preserving** constraints. Existing works either use hand-crafted triggers (e.g., Motif, LIA) or generate triggers without modeling stealth properties (e.g., GTA), which leads to high detectability and limited robustness.
>
> In **Appendix C.1**, we theoretically show that graph-level attacks cause larger distributional shifts than node-level ones, making stealth significantly harder. This arises from two challenges:
>
> - **Structural deviation**: Rare subgraph triggers diverge from clean graph distributions and are easily flagged by detectors;
> - **Semantic deviation**: Label flipping in non-clean-label settings breaks the consistency between structure and label, further increasing anomaly scores.
>
> These highlight a core dilemma: high-ASR triggers are often less stealthy, and stealthier ones tend to be ineffective.
>
> To address this, we propose **DPSBA**, which is (to our knowledge) the first to integrate:
> 1. A **clean-label pipeline** that avoids semantic inconsistency;
> 2. **Anomaly-aware adversarial training** using distribution-aware discriminators on structure/features;
> 3. A **model-agnostic theoretical bound** (Appendix C.2) linking the total variation distance between clean and poisoned distributions to anomaly detectability (AUC).
>
> Together, these designs enable DPSBA to effectively balance stealth and success—an underexplored but critical problem in graph-level backdoors.
>
> We appreciate the opportunity to clarify this and will improve the emphasis on novelty in the revision.
>
> ---
>
> ## **Response to Comments on Experimental Scope and Generalizability (Weaknesses – Experiment & Final Two Questions)**
>
> We sincerely thank the reviewer for raising the important concern about the generalizability of our approach.
>
> Graph learning tasks differ significantly in scope: **node-level** attacks inject localized triggers to misclassify specific nodes by exploiting GNN neighborhood propagation; **link prediction** (edge-level) backdoors manipulate the presence or absence of edges between node pairs in local substructures. In contrast, **graph classification** requires modifying the **global semantics** of an entire graph. This inherently makes stealthy backdoor injection far more challenging, as the trigger must manipulate full-graph embeddings, often involving large or structurally rare subgraphs.
>
> Our theoretical analysis (Appendix C.1) further confirms this: graph-level backdoor attacks induce inherently larger distributional shifts than node-level ones, making **distribution-preserving designs** both essential and under-explored.
>
> While our current focus is on graph classification, we agree that extending DPSBA to node classification or link prediction is a meaningful direction. However, this would require task-specific adjustments to trigger semantics and stealth modeling. We appreciate the reviewer’s suggestion and consider this an exciting avenue for future exploration.
>
> Regarding dataset diversity, our original experiments already cover heterogeneous domains:
> - **PROTEINS_full** – protein graphs for function prediction,
> - **AIDS** – molecular graphs relevant to AIDS drug discovery, and
> - **FRANKENSTEIN** – compound property graphs integrating BURS and MNIST features.
>
> Following the reviewer’s suggestion, we additionally evaluated our method on the **ENZYMES** dataset (a 6-class biomolecular classification task), which is finer-grained and more reflective of real-world scenarios. As shown below, **DPSBA consistently outperforms** all baselines in terms of both attack success rate (ASR) and stealth (CAD, AUC) across three surrogate models. Notably, most baselines (except GTA) fail on this dataset due to high attribute and structure variability, underscoring the advantage of our distribution-preserving and adaptive design.
>
> | Dataset  | Surrogate Model | Metric | ER-B  | LIA   | GTA   | Motif | Motif-S | Ours  |
> |----------|------------------|--------|-------|-------|-------|-------|---------|-------|
> | ENZYMES  | GCN              | ASR    | 26.09 | 30.43 | 95.33 | 21.74 | 15.21   | **96.67** |
> |          |                  | CAD    | 4.17  | 4.99  | 3.00  | 4.99  | **-1.67**   | -0.67 |
> |          |                  | AUC    | 68.32 | 66.15 | 71.20 | 71.35 | 66.22   | **66.11** |
> |          | GIN              | ASR    | 37.83 | 27.02 | 96.00 | 16.21 | 12.16   | **99.33** |
> |          |                  | CAD    | 9.17  | 10.00 | 2.67  | 8.33  | 4.17    | **-0.33**|
> |          |                  | AUC    | 71.40 | 62.01 | 76.42 | 68.18 | 65.78   | **41.20**|
> |          | SAGPool          | ASR    | 29.54 | 38.63 | 100.0 | 15.91 | 11.37   | **100.0** |
> |          |                  | CAD    | 4.33  | 6.67  | 5.00  | 10.83 | **3.33**    | 4.00  |
> |          |                  | AUC    | 57.73 | 63.98 | 70.37 | 75.47 | 69.48   | **49.41** |
>
> We will include these results and the corresponding analysis in the revised paper.
>
> ---
>
> ## **Response to Comments on Poisoning Rate and Trigger Magnitude Sensitivity (Final Two Weaknesses & Question 3)**
>
> We appreciate the reviewer’s insightful suggestion regarding the evaluation of poisoning rate and trigger magnitude. To address this, we have conducted a comprehensive sensitivity analysis (Appendix E.5 and E.6) where both the surrogate and victim models are GCNs:
>
> - **Figure E6** investigates the effect of varying poisoning rate on ASR, CAD, and AUC across all datasets.
> - **Figure E7** explores the impact of trigger magnitude (i.e., number of injected nodes $M$) on attack performance.
>
> These experiments demonstrate that **DPSBA remains effective and stealthy across a wide range of poisoning budgets and trigger sizes**, showing stable ASR with minimal CAD increase, even under low-resource settings. We will highlight these results more clearly in the main text and provide extended analysis for additional surrogate models.
>
> In particular, to respond to the reviewer’s question on achieving 98–99% ASR on PROTEINS\_full, we increased the poisoning rate progressively. We find that a **15% poisoning rate** is required to exceed 99% ASR in this dataset. We attribute this to the **complex structural distribution of PROTEINS\_full**, which has a **node-to-edge ratio of 1:1.86**, compared to the approximately 1:1 ratio in AIDS and FRANKENSTEIN. This makes effective and stealthy manipulation inherently more difficult, yet **DPSBA still achieves top performance**.
>
> The results are summarized below:
>
> | Dataset        | Model | Metric | Poison Rate 9% | Poison Rate 11% | Poison Rate 13% | Poison Rate 15% |
> |----------------|--------|--------|----------------|----------------|----------------|----------------|
> | PROTEINS\_full | GCN    | ASR    | 83.15          | 89.61          | 94.98          | 99.28       |
> |                |        | CAD    | 9.59           | 10.30          | 11.85          | 13.18     |
>
> We thank the reviewer again for encouraging deeper empirical evaluation. These results further support the robustness and adaptability of our framework under various threat model assumptions.
>
> ---
>
> ## **Response to Comment on Broader Societal Impacts (Limitations)**
>
> We have addressed this concern in **Appendix G**, where we explicitly discuss the potential misuse of stealthy graph backdoor attacks.

---

### Note · Authors · 2025-08-12

We sincerely thank the AC and all reviewers for their time, effort, and constructive feedback. All four reviewers provided positive assessments, acknowledging the paper’s technical soundness, practical relevance, and clarity.

We have addressed all concerns through detailed clarifications, additional experiments, and expanded discussions on methodology, generalization, transferability, and parameter sensitivity. The revisions resolve the identified issues and further reinforce the paper’s contributions. **We will incorporate the supplementary discussions and experiments into the final version.**

With the positive overall feedback, the resolved concerns, and the demonstrated novelty and impact, we believe the paper is ready for acceptance.

---

### Decision · Program_Chairs · 2025-09-17

**Decision:**

Accept (poster)

**Comment:**

The paper studies backdoor attacks on graph neural networks: the attacker would like to keep the behavior of the learned network similar on most inputs, but wants to control the learner’s predictions for some examples by modifying them stealthily. That is, the attacker wants to insert a back door into the learned model.

In a less common setup compared to past work, the paper studies whole-graph classification, instead of node or link prediction. This is a more difficult case, since small modifications to the input graph might not influence the classifier, and large modifications might be too easy to detect. (In this regard, Figure 1 is convincing: it shows how easily an anomaly detector can foil some previously-proposed attacks.)

The threat model is: at training time, the attacker has no influence over labels (the clean-label setting), but can influence a small fraction of the training examples by inserting a small number of extra nodes and edges into their input graphs. The attacker cannot delete or change any existing nodes or edges in the training graphs. At test time, the attacker can influence a small fraction of test examples, again by inserting extra nodes and edges into the input. The attacker wins if it is able to cause the learned model to predict a given target class on the modified examples, instead of their true class. The attacker loses if it is detected, either at train time or at test time. The attacker has no knowledge about or influence over the training process beyond its limited ability to alter training data: no ability to attack or surveil the computers that run the training loop, and no information about the model or hyperparameters for training. (In practice the stealth condition means (1) the defender runs anomaly detection on its training and test examples, and the attacker needs not to trigger the detector, as well as (2) the attacker wants not to compromise the learner’s accuracy much on unaltered examples.)

The paper presents a few interesting insights. First, difficult training examples of the target class provide a good attack opportunity: the learner has trouble finding the true signal in these examples, and therefore if the attacker provides a tempting enough false signal, the learner will latch onto it. Second, finding a successful attack while avoiding detection is a constrained optimization problem, and so we can solve it as a game between the attacker and defender trying to minimize/maximize a Lagrangian. Third, insights 1 and 2 can still work even if we don’t have an exact model of the learner or detector: we can still find successful attacks by optimizing against a surrogate defender. In addition to the above, the reviewer/author discussion touched on some other insights, which will help make the final version of the paper better. For example, there was a nice discussion of failure modes.

All these insights are instantiated in a system, DPSBA, and tested on some benchmark graph classification datasets. The system achieves a better tradeoff between attack success rate and attack detection rate compared to prior work.

The reviewers agree that the paper is fairly clear to start, and after modifications according to the discussion, will be clearer. The problem setup is interesting and important, and the paper’s new insights add to our understanding of how to solve it. The authors were responsive and helpful during the discussion.